

# Trace elements in PM$_{2.5}$ aerosols in East Asian outflow in the spring of 2018: Emission, transport, and source apportionment

Takuma Miyakawa[1], Akinori Ito[1], Chunmao Zhu[1], Atsushi Shimizu[2], Erika Matsumoto[3], Yusuke Mizuno[3], Yugo Kanaya[1]

[1]Research Institute for Global Change, Japan Agency for Marine-Earth Science and Technology, Yokohama, 236-0001, Japan

[2]Regional Environment Conservation Division, National Institute for Environmental Studies, Tsukuba, 305-8506, Japan
[3]Horiba Ltd., Ohtsu, 520-0102, Japan

*Correspondence to*: Takuma Miyakawa (miyakawat@jamstec.go.jp)

**Abstract.** Trace metals in aerosol particles impact ocean biogeochemistry. Therefore, semi-continuous measurements of the elemental composition of fine mode (PM$_{2.5}$) aerosols were conducted using an automated X-ray fluorescence analyzer on a remote island of Japan during the spring of 2018. The temporal variations of mass concentrations of geochemically important elements for this period, such as S, Pb, Cu, Si, and Fe, and their relationships with emission tracers carbon monoxide (CO) and black carbon (BC), were reported. The Integrated Massively Parallel Atmospheric Chemical Transport (IMPACT) model was used to evaluate the source apportionment of these components and was evaluated in terms of emissions and wet removal processes. Pb and Cu were found to have originated mainly from anthropogenic sources (98% and 93% on average, respectively) over the East Asian continent. A positive correlation of Pb and Cu with BC and CO was found during the study period, indicating that the emission sources of these metals share the region where the large CO (and BC) emission sources are located. The air masses with minimized impacts of the wet removal during the transport were extracted to elucidate the emission ratio of Pb and Cu to CO, which were, for the first time, evaluated as 152.7 and 63.1 µg g$^{-1}$, respectively, during the spring of 2018 in the East Asian outflow. The analysis of the tagged tracer simulations by the IMPACT model confirmed that BC and Si can be used as tracers for anthropogenic and dust emissions, respectively, during the observation period. The source apportionment of Fe and Mn in PM$_{2.5}$ aerosols was conducted using Si and BC tracers, which revealed that the anthropogenic contribution was 17% and 44% on average, respectively. Based on the air mass origins of Fe and Mn, their anthropogenic fraction varied from 2% to 29% and 9% to 68%, respectively, during the high PM$_{2.5}$ concentration periods. However, despite minor anthropogenic contributions of Fe, they can adversely affect human health and ocean biogeochemistry owing to their higher water solubility. The modeled BC, Pb, Cu, and Fe were evaluated by separately diagnosing their emission and transport. Ratios of modeled to observed concentrations for these components were analyzed in terms of the accumulated precipitation along the transport from the East Asian continent. The current model simulations were found to overestimate the emissions (based on Community Emissions Data System, CEDS v2021-02-05) of BC by 44% and underestimate Cu by 45%, anthropogenic Fe by 28% in East Asia, and the wet deposition rates for BC and Pb. Overall, Cu in East Asia exhibited a different nature from BC and Pb in terms of emission sources and wet removal.



# 1 Introduction

Aerosol particles play essential roles in the atmosphere, as they affect Earth's radiative balance (Szopa et al., 2021 and references therein), human health (e.g., Pope and Dockery, 2006), and ocean biogeochemistry (e.g., Tagliabue et al., 2017). The significance of metallic elements in aerosol particles is in debate, even though their contribution to total aerosol masses or aerosol optical depth is not always dominant in a wide variety of atmospheric environments (Myhre et al., 2013). Recent studies suggest that airborne anthropogenic iron oxides ($FeO_X$) (e.g., magnetite) particles efficiently absorb solar radiation and pose a significant impact on the regional radiative forcing (Ito et al., 2021; Lamb et al., 2021; Moteki et al., 2017). Transition metals in fine particles such as Cu, Mn, and Fe have strong impacts on the formation of reactive oxygen species (ROS) in cells (e.g., Charrier and Anastasio, 2012). The oxidative potential of aerosol particles, which have the potential to generate ROS in cells and cause oxidative stress to cells, has been concerned. Using the mass concentrations of the $PM_{2.5}$ aerosols (particulate matter with an aerodynamic diameter smaller than 2.5 μm) obtained from more than 60 sites worldwide, Lakey et al. (2016) demonstrated a model simulation of ROS production in the epithelial lining fluid (ELF) and showed that the ROS concentrations in the ELF increase with the increase in the $PM_{2.5}$ concentrations. They further suggested that the ROS concentrations greatly vary at the same levels as the $PM_{2.5}$ concentrations and that this variability is attributed to the constituents of the $PM_{2.5}$ aerosols, such as transition metals and organics. In the ocean biogeochemical cycles, Fe and Mn play a role as growth-limiting nutrient for the marine phytoplankton in the high nitrate, low chlorophyll (HNLC) regions such as the northwestern Pacific Ocean and Southern Ocean, and the leachable fraction of Fe is a key to assessing the bioaccessibility of Fe in the surface seawater (Mahowald et al., 2018 and references therein). Some other trace metals can limit microbial biogeochemistry (e.g., Zn, Co, and Mn; Moore et al., 2013), and their higher concentrations can be toxic to some plankton (e.g., Cu; Jordi et al., 2012; Yang et al., 2019). This indicates that atmospheric depositions of these trace metals onto the ocean surface can alter the community composition and potentially inhibit biological productivity.

Asia is recognized as one of the massive anthropogenic emission regions of metallic aerosols (Pacyna and Pacyna, 2001). The concentration levels of trace metals in the East Asian outflow have been characterized in some islands facing the Sea of Japan during 1988–1990 (Mukai and Suzuki, 1996) and East China Sea during 2012–2014 (Shimada et al., 2018). These studies suggested that the metal concentrations are elevated in the air masses transported from the continent, while they are lowest in the summer when the background air masses are frequently advected from the Pacific Ocean. Mukai and Suzuki (1996) used monthly mean data sets of almost three years (1988–1990) and reported the importance of the effects of precipitation on the transboundary transport of metallic elements, especially in fine mode aerosols, through the wet removal processes. They did not rigorously investigate the impacts of the wet removal along the transport from the source regions due to technical data limitations (e.g., longer-time filter sampling and the following chemical analyses). The wet removal of black carbon (BC) aerosols originating from combustion sources has been investigated using high-temporal resolution measurements of BC and carbon monoxide (CO), which is inert against wet removal processes and thereby considered a reliable tracer for the combustion sources (Choi et al., 2020; Kanaya et al., 2016; Verma et al., 2011). They suggested that the transport efficiency



of BC from the continent to the outflow varied according to the air mass histories (e.g., transport altitude below or above the cloud, strength of the precipitation during the transport, etc.). A recent study revealed the decadal decreasing trends (2009–2019) of total $PM_{2.5}$ and BC aerosol concentrations in East Asian outflow, which can be attributed to the rapid reduction of anthropogenic emissions in China (Kanaya et al., 2020). Since atmospheric concentrations of metallic aerosols emitted from

combustion sources are expected to be reduced, further studies are urgently needed on recent concentration levels with high temporal resolution (e.g., Fang et al., 2015; Liu et al., 2019) and their atmospheric behaviors such as transport and removal (e.g., Yoshida et al., 2020) to understand the responses of metallic aerosols in East Asia to the emission regulations and predict future climate change affecting atmospheric transport.

Modeling attempts for the simulation of the multi-elemental compositions of atmospheric aerosols have been challenging.

Only limited studies have tackled this issue, especially in East Asia (Chatani et al., 2021; Kajino et al., 2020; 2021). The Integrated Massively Parallel Atmospheric Chemical Transport Model (IMPACT; e.g., Ito et al., 2023) has the capability to accurately simulate the concentrations of Fe aerosols and the variations in its bioaccessibility by considering the chemical aging process. Recently, the IMPACT model was refined to simulate the multi-elemental compositions of atmospheric aerosols, and the IMPACT model was evaluated in terms of its global scale applicability in a previous study (Ito and Miyakawa, 2023).

The conventional way to accurately quantify the elemental compositions, for example, by Inductively-coupled plasma mass spectrometry (ICP-MS), requires large-volume air sampling and time- and cost-consuming destructive pretreatment steps for the analyses. Unlike ICP-MS, the X-ray fluorescence (XRF) technique has the benefit of easily conducting non-destructive analysis of many samples. Continuous Particulate Monitor with the XRF analysis (PX-375) has been developed by Horiba Ltd. (Asano et al., 2017). In this study, to investigate the emission and transport of trace metals in $PM_{2.5}$ aerosols from the East

Asian continent, PX-375 was deployed on Fukue Island (Kanaya et al., 2016; 2020; Miyakawa et al., 2017; 2019) in the spring (March–April–May) of 2018. Furthermore, collocated measurements of BC and CO concentrations (Kanaya et al., 2016) were integrated into the detailed data analyses to investigate the emission and transport of targeted metallic elements. The IMPACT model was evaluated in terms of emission and transport of the targeted metallic elements in $PM_{2.5}$ aerosols, and was used to investigate the source apportionment of metals in East Asian outflow based on tagged tracer simulation results.

**2 Material and methods**

**2.1 Observations**

Continuous measurement of trace gases and $PM_{2.5}$ aerosols has been conducted at Fukue Island, which is a remote island in western Japan, since February 2009 (Kanaya et al., 2016). The observation site was located at the Fukue Island Atmospheric Environment Monitoring Station (Fukue site; 32.75° N, 128.68° E). The mixing ratio of CO was measured using a

nondispersive infrared (NDIR) CO monitor (model 48C, Thermo Scientific, Inc., USA), and mass concentrations of fine particles ($PM_{2.5}$) and BC were measured using a Synchronized Hybrid Ambient Real-time Particulate (SHARP) monitor (model 5030, Thermo Scientific, Inc., USA) and Multi Angle Absorption Photometer (MAAP, model 5012, Thermo Scientific,



Inc., USA), respectively. According to the intercomparison studies between MAAP and other independent instruments (Kanaya et al., 2013; 2016; Miyakawa et al., 2019)), the mass absorption cross section for the MAAP was modified from the

manufacturer's recommendation of 6.6 to 10.3 $m^2 g^{-1}$. The BC observations using the MAAP were evaluated by a Single Particle Soot Photometer (SP2; Droplet Measurement Technologies, Longmont, Colorado, USA) (Miyakawa et al., 2019). Details of the CO and $PM_{2.5}$ measurements are outlined in Kanaya et al. (2016; 2020), some of which revealed that the elevated concentrations of CO, BC, and $PM_{2.5}$ at the observation site were mainly affected by the transboundary transport of polluted air masses from the Asian continent (i.e., long-range transport) with negligible impact of local emission sources (e.g., Kanaya

et al., 2016).

Elemental compositions of $PM_{2.5}$ aerosols were measured using a Continuous Particulate Monitor with the XRF analysis (PX-375, Horiba Ltd., Kyoto, Japan). The instrumental design is described in Asano et al. (2017). However, the instrument can be briefly described as follows: PX-375 consists of a collection unit for aerosol particles on a filter, the measurement unit for the mass of the collected aerosol particles, and an XRF analysis chamber; the filter tape used for the particle collection was

fabricated by Horiba Ltd. (TFH-01, Horiba Ltd.); and TFH-01 is a polytetrafluoroethylene (PTFE) filter with non-woven fabric polyethylene (PE) and polyethylene terephthalate (PET) backing, which was designed to mechanically strengthen the filter structure. The total mass of $PM_{2.5}$ aerosols collected on the filter was analyzed using a radiocarbon-based beta-ray attenuation method during the particle collection. The 4 hourly $PM_{2.5}$ mass concentrations measured using the PX-375 were compared with those measured using the SHARP monitor. PX-375-derived $PM_{2.5}$ mass concentrations were found to be in

good agreement (within 10%) with those observed by the SHARP monitor. After sampling, a particle-laden spot on the filter was transferred into the XRF analysis chamber by advancing the filter roll tape and then analyzed by the XRF technique. The ambient air was drawn at 16.7 liters per minute through the $PM_{2.5}$ cyclone (URG-2000-30EH, URG Corp., North Carolina, USA) into the PX-375 instrument. During the observation period, the $PM_{2.5}$ aerosol particles were collected for 4 h, and the XRF analyses were performed with 4000 sec of X-ray irradiation at 15 and 50 kV. The typical volume of the sampled air was

4 $m^3$ per particle-laden spot. The limits of detection (LODs) were evaluated by repeated measurements placing the high efficiency particulate air (HEPA) filter in front of the PX-375 to introduce particle-free air into the PX-375. The LODs were defined as three times the standard deviations of the measured values by the PX-375 at the observation site and were evaluated to be 1.57, 0.28, 0.70, 0.69, 25.6, 18.3, 1.95, 8.50, and 9.50 ng $m^{-3}$ for Fe, Pb, Cu, Mn, K, Cl, Ca, Si, and S as sulfate, respectively. The particle-laden spots on the filter roll tape were cut into pieces after the observation period and then stored in

the refrigerator (at approximately −20 °C). The particle spots can be reanalyzed by different techniques (e.g., Zhu et al., 2021). In this study, particle spots collected before this observations were reanalyzed for the recalibrations of the onsite measurements using different chemical analysis techniques: ion chromatography (IC) for Cl (as chloride) and S (as sulfate) and ICP-MS for Fe, Pb, Mn, Cu, Mn, K, and Ca. Considering the intercomparison results, the uncertainties in the on-site measurements of the concentrations of Cl, S, Fe, Pb, Mn, Cu, K, and Ca were evaluated to be 66%, 6%, 26%, 15%, 23%, 30%, 28%, and 33%,

respectively.



Lidar-derived aerosol extinction coefficients at 532 nm (e.g., Shimizu et al., 2015) were analyzed to investigate the impacts of Asian dust on the measured aerosol concentrations. The lidar measurements at the same site were part of the lidar network over Asia and are maintained by the National Institute for Environmental Studies in Japan (Shimizu et al., 2004; Sugimoto et al., 2003). The contribution of dust to the lidar-derived aerosol extinction coefficient was estimated using the aerosol

depolarization ratio, assuming that dust particles are externally mixed with spherical particles. All the lidar data were obtained with 15 min temporal and 30 m vertical resolutions. The dust aerosol extinction coefficients near the surface were estimated by averaging the dust extinction coefficients between heights of 120 and 240 m (the contribution below 120 m was not measured and, thereby, was omitted in this study) and were merged into 4 hourly elemental composition datasets. Note that the lidar measured the particles for the whole size range, including coarse particles (>2.5 µm in diameter), unlike the in-situ

measurements using the PX-375.

**2.2 Meteorological data and air mass origin analyses**

Backward trajectories were calculated from the observation site to elucidate the impact of Asian outflow. Five day backward trajectories from the observation site (at a starting altitude of 0.5, 1.0, and 1.5 km) were calculated every hour using the NOAA Hybrid Single-Particle Lagrangian Integrated Trajectory Model (Draxler and Rolph, 2012; Rolph, 2012; Stein et al., 2015)

with the meteorological data sets with a resolution of 1° in latitude and longitude (NCEP's GDAS). The accumulated precipitation along trajectories (APT) for three days before the measurement was calculated to discuss the wet removal during transport (Kanaya et al., 2016; Miyakawa et al., 2017). The concentration-weighted trajectory (CWT) for a pollutant in a particular grid of latitude and longitude (i, j) was measured to determine the pollutant source strength in a grid of a receptor site. In this study, this analysis was applied to the four hourly aerosol mass concentrations observed at the observation site.

The CWT for the aerosol concentrations was determined as follows (Kabashnikov et al., 2011):

$$CWT_{i,j} = \dfrac{\sum_{k=1}^{L}(C_k \cdot \tau_{i,j,k})}{\sum_{k=1}^{L} \tau_{i,j,k}} \qquad \text{(Eq. 1)}$$

where $C_k$ is the mass concentrations of total and selected contents in $PM_{2.5}$ aerosols corresponding to the arrival of backward

trajectory k; $\tau_{i,j,k}$ is the residence time of trajectory segment endpoints in the grid cell (latitude i and longitude j) for backward trajectory k; and L is the total number of backward trajectories (6600).

Since the 4 hourly concentrations are discussed in this study, the same values of $C_k$ for hourly-calculated backward trajectories for a 4 hour duration were used for calculating the CWT. The resolution of a grid cell was 1° in latitude and longitude. $\tau_{i,j,k}$ was determined by counting the number of the calculated hourly trajectory segment endpoints only below 2 km altitude in

each grid cell for each trajectory. The reason for limiting the altitude for counting backward trajectories to below 2 km was to investigate the impact of the surface emission sources over the land area.



**2.3 Integrated Massively Parallel Atmospheric Chemical Transport Model**

In this study, IMPACT model (Ito and Miyakawa, 2023; and references therein) was deployed to simulate three-dimensional distribution of the atmospheric composition including targeted aerosol species, such as Fe. Ito and Miyakawa (2023) modified
the IMPACT model to expand its capability to simulate various elements other than Fe, including Pb, Cu, Mn, and Si. They improved the IMPACT model to include metal smelting as an emission source of Fe, which has not well been evaluated in previous studies. The simulated atmospheric constituents of CO, BC, sulfate, Pb, Cu, Mn, and Fe concentrations were compared with the observations of this study. Source apportionment of trace elements in $PM_{2.5}$ aerosols was also analyzed based on the tagged tracer simulations using the IMPACT model, which has a capability to separately simulate lithogenic (i.e.,
mineral dust), pyrogenic (i.e., biomass burning), and anthropogenic (combustion and non-combustion) contributions of the selected species such as sulfate, BC, and Fe in $PM_{2.5}$ aerosols. A brief description of the IMPACT model related to BC and the trace metals is as follows:

The model simulations were performed using a horizontal resolution of $2.0° × 2.5°$ (latitude × longitude) and 47 vertical layers. An emission inventory, Community Emissions Data System (CEDS, v-2021-02-05, O'Rourke et al., 2021), was used for fine
particulate matter and BC emitting from anthropogenic sources. The metal content of $PM_{2.5}$ aerosols from lithogenic, pyrogenic, and anthropogenic sources was obtained from the compilation of source-specific aerosol measurements (Ito et al., 2018; Kajino et al., 2020; Reff et al., 2009). Since CEDS does not include the emission of BC (a combustion source tracer) from metal production sector (e.g., the production of iron and steel, aluminum, and other non-ferrous metals), Fe emissions from the metal production sector was estimated by scaling sulfur dioxide ($SO_2$) emission from CEDS based on the relative emissions of Fe to
$SO_2$ (Rathod et al., 2020) in each country.

# 3 Results and discussion

## 3.1 Meteorology

During the spring season, typical meteorological conditions in East Asian outflow were reported in our previous studies (Kanaya et al., 2016; Miyakawa et al., 2017; 2019). The mid-latitude region (35–50° N, 120–140° E) in East Asia is influenced
by a modest monsoonal northwesterly flow from the continent to the observation area, while the subtropical region (20–30° N, 110–130° E) is influenced by a persistent southwesterly flow, part of which converges into the observation area, confluent with the north westerlies from the continent. As the low-level southerly flow transports warm and moist air into the observation area to sustain a large amount of precipitation, the transport efficiency of aerosol particles in these air masses is low due to wet removal. These similar transport patterns were observed in the spring of 2018 (**Figure S1**). From March to May 2018, the air
masses were frequently transported from the East Asian continent to the observation site on Fukue Island, with small Japanese emission impacts (**Figure S2**). The APT had no distinct seasonal progressions in the spring of 2018 and had no correlation





with the residence time over the continent (**Figure S3**). Notably, air masses with and without substantial impacts from wet depositions were observed during the observation period.

### 3.2 Springtime PM$_{2.5}$ aerosols in the East Asian outflow in 2018

The temporal variations of PM$_{2.5}$ aerosols, sulfate, Si, Fe, Ca, Mn, Pb, Cu, BC, and CO concentrations in the spring of 2018 are shown in **Figure 1**. The multi-elemental characterizations of PM$_{2.5}$ aerosols successfully captured the source-specific temporal variations among elements. For example, in the middle of April (April 15–19, 2018), at a strong dust event, no large impact of anthropogenic emissions (no large enhancement of CO, BC, and sulfate) was observed by the PX-375. At the end of March (March 25–26, 2018), BC, CO, Pb, Cu, and sulfate concentrations were found to be enhanced, while dust-related

elements such as Si and Ca showed relatively low concentrations compared to the middle of April, 2018. Furthermore, three selected periods (P1 (March 25–26, 2018), P2 (April 15–18, 2018), and P3 (April 29, 2018)) when the Japanese short-term air quality standard (AQS) for PM$_{2.5}$ was violated (daily concentration > 35 µg m$^{-3}$) are highlighted in this study. 5-d backward trajectories for P1, P2, and P3 are also shown in **Figure 1**. For P1 and P3, large combustion source areas (**Figure S4**) were the air mass pathways, whereas the Gobi Desert could be a possible dust source area for P2. This is consistent with the observed

temporal changes in the concentrations of atmospheric compositions, including anthropogenic combustion tracer (CO and BC), dust tracer (Si), and secondary formation tracer (sulfate). During P2, Si, Ca, Mn, and Fe were abundant in the compositions of dust particles and showed the highest concentrations in the study period.

     The CWTs of PM$_{2.5}$, sulfate, Pb, Cu, BC, and Fe are shown in **Figure 2**. The spatial distributions of CWTs of Pb and Cu were similar to those of BC and emissions of CO (**Figure S4**), indicating that these elements share strong emission regions with CO,

where anthropogenic combustion sources dominate. On the contrary, the CWTs of Fe were found to be clearly different from them. The dust aerosols were found to be transported from the desert regions (e.g., the Gobi Desert) and important for the large enhancement of Fe concentrations in the East Asian outflow.

### 3.3 Outlook on the concentrations and source apportionment of selected components simulated using the IMPACT model

The performance of the IMPACT model predicting the concentrations of selected aerosol components (BC, Fe, Si, Pb, and Cu) in PM$_{2.5}$ aerosols has already been described by Ito and Miyakawa (2023). The normalized mean biases (NMB; Pearson's correlation coefficients, r) were calculated to be 52% (0.74), 11% (0.69), 10% (0.65), −14% (0.79), −51% (0.67), −15% (0.65), and −66% (0.59) for BC, Fe, Si, Pb, Cu, Ca, and Mn, respectively. This indicated that (1) the concentrations of Fe, Si, Pb, and Ca were reasonably well predicted (NMB < ±20%) as their r values were high (0.65–0.79), (2) the concentrations of Cu and

Mn had large negative NMB values (NMB < −50%) with moderately high r values (0.59–0.67), and (3) the concentrations of BC were overestimated but with a high r value (> 0.7).

     The tagged tracer analyses using the IMPACT model simulations suggested that the observed concentrations of BC, Pb, and Cu in Fukue Island were dominated by anthropogenic sources over the East Asian continent (concentration-weighted averages



of 94%, 99%, and 93%, respectively), and Si originated from desert areas over the East Asian continent (concentration-weighted average of 92%) during the study period. Miyakawa et al. (2019) used radiocarbon and molecular marker analyses for the carbonaceous aerosols and found that fossil fuel combustion was the dominant source of BC (approximately 90%) in East Asian outflow in the spring of 2015, which is consistent with the dominant anthropogenic contribution to BC concentrations modeled in this study. Ca in $PM_{2.5}$ aerosols was also primarily affected by dust (concentration-weighted average of 71%), whereas sea sprays contributed secondarily (concentration-weighted average of 17%), which is higher than anthropogenic contributions (concentration-weighted average of 10%). It was found that natural sources were important for the observed Ca concentrations in $PM_{2.5}$ aerosols and that Si is a better tracer for dust aerosols than Ca.

In this study, the emission and transport of the anthropogenic components BC, Pb, and Cu in $PM_{2.5}$ aerosols and the source apportionment of Fe and Mn in $PM_{2.5}$ aerosols were investigated using specific tracers for the anthropogenic and dust emissions via the evaluations of the IMPACT model performances. The former, as discussed in Sect. 3.4, included the evaluations of the emission ratios of Pb and Cu with BC and CO, the wet removals of Pb and Cu, and the comparisons of the observations with the model simulations. The latter, as discussed in Sect. 3.5, included the analyses of the results of the source apportionment in terms of the elemental composition of $PM_{2.5}$ dust aerosols and the enhancement ratios of anthropogenic Fe and Mn to BC in the springtime East Asian outflow.

## 3.4 Emissions and transport of anthropogenic aerosol components (BC, Pb, and Cu)

Pb, Cu, and a combustion tracer, BC, were analyzed in terms of their emission and transport in the East Asian outflow, considering the dominances of their anthropogenic sources based on the IMPACT model results (Sect. 3.3). Pb and Cu showed a positive correlation with BC ($r^2$ = 0.72 and 0.66, respectively; **Figure 3**), which is consistent with their temporal variations (**Figure 1b**). Since BC aerosol particles in the East Asian outflow are efficiently removed from the atmosphere by the rainout process (Kanaya et al., 2016; Miyakawa et al., 2017; Moteki et al., 2012), Pb and Cu can also be affected by the wet depositions during transport. The correlations of Pb and Cu with CO were investigated to examine the removal of Pb and Cu relative to CO. In this study, the enhancement of CO from its background was used for this analysis because CO has a longer lifetime of approximately 2 months than a typical transport time scale from the East Asian continent to Fukue Island (approximately 2d; Kanaya et al., 2016) and does not get affected by wet deposition processes. The enhancement of CO from its background $CO_{bkg}$ ($\Delta CO \equiv CO - CO_{bkg}$) was calculated following Kanaya et al. (2016; 2020). **Figures 3c** and **3d** depict the correlations of Pb and Cu with $\Delta CO$, respectively, and show that Pb has a better correlation with BC and CO than Cu. This indicates that Cu has different emission sources from BC, Pb, and CO in East Asia, even though the similarity of their geospatial patterns was indicated by the CWTs for BC, Pb, and Cu. The datasets of Pb and Cu were classified into APT > 1 mm h and ≤ 1 mm h (i.e., more and less affected by the wet removal, respectively) to roughly diagnose the effects of wet removal on the transport of Pb and Cu (**Figures 3c** and **3d**). Notably, the relative enhancement of Pb to $\Delta CO$ was smaller for the APT > 1 mm h than for that ≤ 1mm h.





To compensate for the concentration changes induced by the atmospheric dilution during the transport, the concentrations of X (X = BC, Pb, and Cu) normalized by the longer-lived species concentrations were analyzed to diagnose the transport efficiency. The enhancement ratios of component X to CO, $\Delta X/\Delta CO$, during the observation period were evaluated by selecting the data with an APT of zero (**Table 1**). The background levels for BC, Pb, and Cu were assumed to be zero (Kanaya et al., 2016; Miyakawa et al., 2017). The emission ratios of X to CO ($ER_{XtoCO}$) were determined by selecting the data points of $\Delta X/\Delta CO$ when $\Delta CO$ was > 20 ppb, continental air masses were transported, concentrations of Pb and Cu were higher than their LOD, and APT equals zero (i.e., no precipitation during the transport), which was basically the same as the method given by Kanaya et al. (2016). The $ER_{XtoCO}$ values for X = BC, Pb, and Cu during the entire observation period (March, April, and May; MAM) were evaluated to be 4.75 ($\pm$1.39) ng m$^{-3}$ ppb$^{-1}$, 177.8 ($\pm$88.8) pg m$^{-3}$ ppb$^{-1}$, and 73.5 ($\pm$42.2) pg m$^{-3}$ ppb$^{-1}$, respectively. Since no significant difference (p > 0.05) was found in the $ER_{XtoCO}$ for BC, Pb, and Cu between April and May in 2018, they were merged into one dataset group "April–May" in 2018. The $ER_{XtoCO}$ for Pb and Cu had relatively large and significant differences (p < 0.01) between March and April–May in 2018 compared to that for BC, indicating the seasonal variations in emission ratios of the anthropogenic sources of Pb and Cu over the East Asian continent. To the best of our knowledge, this is the first attempt to evaluate the emission ratios of Pb and Cu to CO in East Asia.

**Table 1** $ER_{XtoCO}$ in the East Asian outflow.

| Month | $ER_{BCtoCO}$ [ng m$^{-3}$ ppb$^{-1}$] | | | $ER_{PbtoCO}$ [pg m$^{-3}$ ppb$^{-1}$] | | | $ER_{CutoCO}$ [pg m$^{-3}$ ppb$^{-1}$] | | |
|---|---|---|---|---|---|---|---|---|---|
| | Avg. | Std. | N | Avg. | Std. | N | Avg. | Std. | N |
| March | 4.22 | 1.04 | 42 | 157.6 | 82.0 | 42 | 46.7 | 19.7 | 42 |
| April | 5.27 | 1.53 | 42 | 190.2 | 89.6 | 42 | 94.7 | 40.8 | 41 |
| May | 4.94 | 1.13 | 4 | 258.4 | 76.5 | 4 | 137.0 | 43.4 | 4 |
| Apr–May | 5.24 | 1.49 | 46 | 196.1 | 91.7 | 46 | 98.5 | 42.3 | 45 |
| MAM* | 4.75 | 1.39 | 88 | 177.8 | 88.8 | 88 | 73.5 | 42.2 | 87 |

*MAM indicates March, April, and May of 2018 (i.e., the entire observation period).

$ER_{XtoCO}$ values were used to calculate the transport efficiency of X ($TE_X$) as ($\Delta X/\Delta CO$)/$ER_{XtoCO}$. **Figure 4** depicts $TE_{BC}$, $TE_{Pb}$, and $TE_{Cu}$ as functions of the APT. As a reference, the long-term mean $TE_{BC}$-APT relationship (2009–2015) evaluated by Kanaya et al. (2016) was compared. While $TE_{Pb}$ showed a similar relationship with the APT to $TE_{BC}$, the decreasing trend of the $TE_{Cu}$-APT relationship was not steep, probably because Cu has characteristics of emission and mixing states different from BC and Pb. Indeed, Kinase et al. (2022) reported continuous observations of trace metals in PM$_{2.5}$ aerosols in an urban area of western Japan and showed that Cu concentrations were poorly correlated (r$^2$ = 0.17) with Pb concentrations when the local impact was significant and that approximately 60% of Cu-containing particles did not contain Pb as analyzed using an electron




microscopic technique. Since the length of the observation periods and number of data points were limited, the development of robust fit functions was difficult in this study. $TE_{Pb}$ decreased with an increase in APT, similar to $TE_{BC}$. Although the tendency of $TE_{Cu}$ was not as steep as that of $TE_{BC}$ and $TE_{Pb}$, its values were significantly less than unity ($p < 0.01$) for the data points with APT values higher than 0.5 mm h. Consequently, the concentrations of BC, Pb, and Cu in $PM_{2.5}$ aerosols can be

concluded to be affected by the wet removal during transport in the East Asian outflow in the spring of 2018.

Modeled BC, Pb, and Cu concentrations were evaluated in terms of APT (**Figures 5** and **S5**). The IMPACT model reasonably predicted the concentrations of Pb and overestimated those of BC by 44% for the data with APT < 1 mm (i.e., small impact of the wet removal during the transport). For BC and Pb, the ratios of modeled to observed concentrations (M/O) with wet removal impacts were higher than those with no precipitation. This result and the observed decreasing trend of $TE_{BC}$ and $TE_{Pb}$ with

increasing APT indicate that the wet removal processes for BC and Pb were weaker in the IMPACT model than in reality. The IMPACT model systematically underestimated the concentrations of Cu by 45% on average, indicating that the anthropogenic Cu emission inventory used in this study needs revision. The trend of M/O ratios against APT for Cu showed no substantial decreasing trend, similar to BC and Pb. The variability of M/O ratios for Cu was not controlled solely by the uncertainty in relation to the wet removal process, which is consistent with the gently decreasing trend of the $TE_{Cu}$-APT relationship.

Nonetheless, the removal processes and emissions of Cu were not properly simulated by the IMPACT model, as indicated by the correlation of the M/O ratios for Cu with $TE_{Cu}$, as shown in **Figure S6**. Both the emissions and transport of Cu aerosols need further investigation in the future.

### 3.5 Source apportionment analyses

### 3.5.1 Validation of multiple linear regression analysis for source apportionment

Based on the IMPACT model simulations, Si and BC can be used as excellent tracers for dust and anthropogenic primary emission sources, respectively; therefore, the source apportionment of Fe and Mn in $PM_{2.5}$ aerosols can be performed using these tracers. Source apportionment of $PM_{2.5}$ aerosols was conducted using BC, Si, and sulfate concentrations ($[BC]_t$, $[Si]_t$, and $[SO_4^{2-}]_t$, respectively) to validate the use of these tracers for the source apportionment of Fe and Mn. Since BC showed a positive correlation with sulfate (**Figure S7**), the temporal variations of BC may not be completely independent from those of

sulfate. However, BC showed a better correlation with an independent combustion source tracer of CO than sulfate (**Figure S7**), indicating that the temporal variations of BC concentrations can account for those of both anthropogenic aerosols and a part of secondary aerosol concentrations. Additionally, sulfate concentrations can be used to constrain the secondary formation component in $PM_{2.5}$ aerosols through the aqueous phase reactions inside aerosol particles and cloud droplets, which cannot be represented by the temporal variations of BC concentrations. Consequently, it can be concluded that both BC and sulfate can

be used for the multiple linear regression (MLR) analysis. The validity of selecting Si, BC, and $SO_4^{2-}$ as the input variables for the MLR models is described in Sect. S1 of the supporting information. The observed temporal variations of $PM_{2.5}$ aerosol concentrations ($[PM_{2.5}]_t$) were fitted in the following MLR model:



$$[PM_{2.5}]_t = g_{BC} \cdot [BC]_t + g_{Si} \cdot [Si]_t + g_{sulfate} \cdot [SO_4^{2-}]_t + C \qquad \text{(Eq. 2)}$$


The coefficients $g_X$ (X = BC, Si, and sulfate) and C (i.e., constant term) were determined by the least squares method. As the coefficient of determination of the correlation between reconstructed and observed $PM_{2.5}$ concentrations was 0.85 (**Figure S9**), this linear combination model accounts for most of the observed variances in $PM_{2.5}$ concentrations. C was evaluated to be 4.95 (±0.60) µg m$^{-3}$, which can correspond to the background level of $PM_{2.5}$ concentration (including sea-salt aerosols) in the East

Asian outflow in the spring of 2018. **Figure 6a** depicts the temporal variations of the observed and source-classified concentrations of $PM_{2.5}$ aerosols during the observation period. The concentrations and contributions of BC-, sulfate-, and dust-related aerosols in $PM_{2.5}$ aerosols for the entire and selected periods are shown in **Figure 6b**. The highest concentrations and dominant contributions of dust-related aerosols were observed during P2, which is consistent with the air mass transport presented in Sect. 3.2. An independent method, co-located Lidar measurement at the same site, was applied to validate our

source apportionment results for the dust contribution (**Figures 6c** and **6d**). Lidar-derived dust contribution to total extinction near the surface was positively correlated with the estimated dust contributions to total $PM_{2.5}$ (i.e., $g_{Si} \cdot [Si]_t / [PM_{2.5}]_t$) for the selected periods (P1, P2, and P3) at higher aerosol loadings during the observation periods. These considerations indicate that this approach can successfully analyze the source apportionment of $PM_{2.5}$ aerosols, especially dust and non-dust contributions, using the temporal variations of tracer compounds.

**3.5.2 Source apportionment of Fe and Mn**

Since the estimation of the dust contribution using Si as a tracer was validated in Sect. 3.5.1, the same approach was applied to the observation-based source apportionment of Fe and Mn. The IMPACT model simulations suggested that the Fe and Mn concentrations at the observation site were affected by anthropogenic and lithogenic sources, with negligible impact from biomass burning during the spring of 2018 (Ito and Miyakawa, 2023). The Eq. (2) for the source apportionment of $PM_{2.5}$ was

modified into the following Eq. (3):

$$[Y]_t = g_{BC,Y} \cdot [BC]_t + g_{Si,Y} \cdot [Si]_t + C_Y \qquad \text{(Eq. 3)}$$

$g_{X,Y}$ (X = BC and Si; Y = Fe and Mn) and $C_Y$ (constant term) were determined by the least squares method. Notably, the BC-

related components of Fe and Mn can be regarded as their anthropogenic contributions due to the lack of secondary formation processes for Fe and Mn. $C_{Fe}$ and $C_{Mn}$ were almost zero in this study. As the coefficients of determination of the correlation between reconstructed and observed Fe and Mn concentrations were 0.97 and 0.92, respectively **(Figure S9)**, this linear combination model also accounts for almost all the observed variances of Fe and Mn concentrations. The anthropogenic contribution to total Fe in $PM_{2.5}$ aerosols was 17% (±11%) on average, which was similar to that simulated by the IMPACT

model (22±17%). Furthermore, the derived range was comparable to that (10–50%) estimated using the stable isotope ratio of



Fe ($\delta^{56}$Fe) of PM$_{2.5}$ aerosols transported from the East Asian continent to the northwestern Pacific Ocean (Kurisu et al., 2021). The anthropogenic contribution to total Mn in PM$_{2.5}$ aerosols was 44% (±20%) on average, which was in reasonable agreement with that simulated by the IMPACT model (42±23%) and was higher than that of Fe. Variations in the anthropogenic and dust contributions to total Fe and Mn concentrations in PM$_{2.5}$ aerosols are further discussed in the following section.

The derived anthropogenic- and dust-Fe concentrations showed reasonable agreements with those simulated by the IMPACT model (most of the data points fell into the range of the factor of 2, **Figure S10**). The M/O ratios for both anthropogenic- and dust-Fe with no precipitation (APT = 0) were approximately 0.7, which was lower than expected from the M/O ratios for total Fe for all the data with or without the impacts of the precipitation as indicated in Sect. 3.3. Since the deviation was approximately 30%, which can be accounted for by the observation uncertainty of Fe, Si, and BC, both anthropogenic Fe and

dust emissions and Fe concentration in dust aerosol particles were underestimated in the model simulations. The derived anthropogenic and dust-Mn concentrations were systematically higher than those simulated by the IMPACT model (**Figure S10**), suggesting that both anthropogenic- and dust-Mn concentrations were underestimated by the IMPACT model.

### 3.5.3 Enhancement ratios of anthropogenic Fe and Mn to BC in the springtime East Asian outflow

$g_{BC,Fe}$ represents the enhancement ratio of anthropogenic-Fe to BC and was determined to be 0.13 (±0.03) µg µg$^{-1}$. This value

is significantly higher than the Fe/BC ratios derived from biomass burning (0.02 µg µg$^{-1}$) observed for fine-mode aerosols in the Amazon (Luo et al., 2008), which is consistent with the IMPACT's model prediction of a negligible influence of biomass burning on the observed Fe concentrations. On the other hand, the derived value here is lower than the magnetite-Fe/BC ratio of 0.25 observed at the same observation site in the spring of 2020 (Yoshida et al., 2020). The contribution of magnetite-Fe to total anthropogenic Fe was evaluated to be approximately 40% (Matsui et al., 2018; Ito and Miyakawa, 2023), indicating a

large difference in anthropogenic-Fe/BC ratios between this and the previous study (Yoshida et al., 2020). The possible reasons for this may be the differences in the year of the observations (2018 vs. 2020), analytical technique, and sampling condition. In the recent decade, BC emissions from China substantially decreased (Kanaya et al., 2020; 2021), indicating that anthropogenic emissions of primary aerosols in China also decreased. However, if the anthropogenic-Fe emissions are not majorly reduced over time, the ratio of anthropogenic-Fe to BC might increase. Since the recent reduction of BC emissions

was primarily attributed to the industrial sector and then the residential sector (Kanaya et al., 2020), the emission changes cannot account for all the differences in the ratio of anthropogenic-Fe to BC. Yoshida et al. (2020) applied the laser-induced incandescence technique to detect airborne single particles containing iron oxide in the size range of 170−2100 nm of mass-equivalent diameter, whereas this study applied the XRF analysis to the bulk aerosol samples collected through the PM$_{2.5}$ cyclone. As concluding the major factor accounting for the discrepancy is difficult, integrated studies applying other

independent analytical techniques are needed in the future. Chemical speciation of Fe-containing aerosols based on X-ray absorption fine structure spectroscopy (Kurisu et al., 2019; 2021; Sakata et al., 2022; Takahashi et al., 2011) could be promising for further investigating the anthropogenic-Fe in the East Asian outflow.



$g_{BC,Mn}$ represents the enhancement ratio of anthropogenic-Mn to BC and was determined to be 17.1 (±1.1) ng µg⁻¹. To the best of our knowledge, this is the first time that the enhancement ratios of Mn to BC are evaluated from the ambient measurements

of anthropogenic sources in the East Asian outflow, which can be used as a useful constraint to estimate the anthropogenic contribution to Mn in East Asian outflow regions.

### 3.5.4 Elemental concentrations of Si, Fe, and Mn in PM₂.₅ dust aerosols

Based on the source apportionment results, elemental concentrations of Si, Fe, and Mn were estimated and compared with those evaluated in previous studies that analyzed Asian dust samples collected at Andong, Deokjeok Island, and Seoul in Korea

(Jeong, 2020) and Yulin in China (Wang et al., 2011) and those simulated by the IMPACT model (Ito and Miyakawa, 2023) (**Figure S11**). $g_{Si}^{-1}$ (= $[Si]_t/(g_{Si} \cdot [Si]_t)$) can be regarded as the concentration of Si in PM₂.₅ dust aerosols and was determined to be 21.6% (20.4–23.1%) in this study. This value is consistent with that given in a previous study (26.99±1.24%) that analyzed Asian dust in total suspended particulate matter (TSP) collected from Korea (Jeong, 2020). Note that 22.1% of PM₂.₅ dust aerosols was prescribed for Si content in the model for comparison with the observations (Ito and Miyakawa, 2023). Although

the uncertainty in the quantification of $[Si]_t$ was not evaluated prior to the observation, the derived values of $g_{Si}^{-1}$ were reasonably comparable to the other independent evaluations. On changing the $[Si]_t$ by ±20%, the values of $g_{Si}^{-1}$ were determined to be 17.3–26.0% (−20% to +20% cases), without significant changes in the fitted values of $g_{BC}$, $g_{sulfate}$, and C. This indicates that the source apportionment of PM₂.₅ aerosol concentrations was not strongly affected by the uncertainty of $[Si]_t$, and the uncertainty of $[Si]_t$ was estimated to be not so high (~15% at most).

$g_{Si,Fe} \cdot g_{Si}^{-1}$ (= $g_{Si,Fe} \cdot [Si]_t/(g_{Si} \cdot [Si]_t)$) represents the concentration of Fe in dust aerosols (7.3±0.5%), which was evaluated in Korea (5.27±0.25%) and China (3.98% of PM₂.₅ aerosols collected at Yulin, China) and was compared with that simulated by the IMPACT model (3.83±0.26% at the observation site). $g_{Si,Mn} \cdot g_{Si}^{-1}$ (= $g_{Si,Mn} \cdot [Si]_t/(g_{Si} \cdot [Si]_t)$) represents the concentration of Mn in dust aerosols (0.21±0.01%), which was evaluated to be 0.12±0.01% by Jeong (2020) and 0.11% by Wang et al. (2011). Note that 0.077% of PM₂.₅ dust aerosols was prescribed for Mn content in the model for comparison with the observations (Ito and

Miyakawa, 2023). The concentrations of Fe and Mn in PM₂.₅ dust aerosols estimated in this study were higher than those given in the independent evaluations. However, the orders of the evaluated elemental concentrations of all the selected elements (Si, Fe, and Mn) were the same as in all the independent evaluations. On changing $[Si]_t$ by ±20%, the derived values of $g_{Si,Fe} \cdot g_{Si}^{-1}$ and $g_{Si,Mn} \cdot g_{Si}^{-1}$ would not change without the significant changes in the determinations of $g_{BC,Fe}$, $g_{BC,Mn}$, $C_{Fe}$, and $C_{Mn}$, because the increases (decreases) in $g_{Si,Fe}$ and $g_{Si,Mn}$ were almost completely compensated by decreasing (increasing) $[Si]_t$. This indicates

that the source apportionment of Fe and Mn aerosol concentrations and their elemental mass concentrations in dust aerosols were not significantly affected by the uncertainty of $[Si]_t$. The uncertainty of $[BC]_t$ (22%) also did not affect the calculations of the fitting coefficients in the MLR model. The possible causes of higher Fe and Mn concentrations may be the contributions of anthropogenic Si-containing particles with higher Fe and Mn contents, such as fly ash (e.g., Li et al., 2012; Liu et al., 2019), and the contributions of "anthropogenic" dust sources associated with the agricultural land use change (~40% in East Asia,

e.g., Ginoux et al., 2012). Due to limited knowledge of the difference in dust elemental composition between natural and





anthropogenic sources, further investigations of the elemental compositions of both natural and anthropogenic dust aerosol particles are needed in the future. Note that the discrepancies can also be partly accounted for by the uncertainties in measuring $[Fe]_t$ and $[Mn]_t$ (26% and 23%, respectively) using the PX-375.

### 3.5.5 Variations in the anthropogenic contribution of total Fe and Mn in PM$_{2.5}$ aerosols

**Figure 7** depicts the temporal variations of the observed and source-classified concentrations of Fe and Mn during the observation period. During P2 (the strong dust period), both elements showed the highest concentrations in the entire observation period. The estimated dust Fe and Mn also showed the highest concentrations in the entire observation period. The anthropogenic sources contributed to 2.1% (±1.2%) and 8.8% (±4.3%) of total Fe and Mn, respectively, in PM$_{2.5}$ aerosols during P2. During P1 and P3, the anthropogenic contributions increased by approximately 30% and 65% of total Fe and Mn,

respectively, in PM$_{2.5}$ aerosols. It was found that depending on the air mass origin, the anthropogenic or dust contributions to the observed Fe and Mn concentrations greatly varied, and the anthropogenic contributions to total Fe and Mn were inversely proportional to the total concentrations of Fe and Mn in the spring of 2018 (**Figure 8**). Since Asian dust aerosols are frequently transported from the desert region (e.g., the Gobi Desert) over the East Asian continent towards the outflow area, such as the Pacific Ocean, in the spring season (e.g., Wan et al., 2020), the large springtime enhancements in the concentrations of Fe and

Mn in the East Asian outflow are substantially affected by the dust emissions over the East Asian continent.

In the East Asian outflow, the fractional solubilities of combustion-derived and dust-derived Fe were estimated to be 11% and 0.9%, respectively (Kurisu et al., 2021). Assuming applying these fractional water solubilities ($f_{sol}$) of Fe to our data, the estimated water-soluble Fe concentration (17.7 ng m$^{-3}$) during P1 and P3 (the periods with fewer impacts of dust) was comparable to 19.0 ng m$^{-3}$ during P2 (the dust-dominated period), even though the total Fe concentrations during P1 and P3

were almost 30% of that during P2. Since the $f_{sol}$ of Fe can be affected by atmospheric processing during transport and can be as high as ~50% for fine mode aerosols over the remote ocean (Sholkovitz et al., 2012 and references therein), the evaluations of water-soluble Fe contributions during P1 and P3 might be substantially underestimated. These considerations supported the importance of anthropogenic sources that deliver water-soluble Fe to the East Asian outflow region, as suggested in previous studies (e.g., Ito and Miyakawa, 2023). The $f_{sol}$ of Mn is typically higher than that of Fe and does not exhibit the same large

variabilities in $f_{sol}$ as Fe (Baker et al., 2013; 2014; Shelley et al., 2018). The enhancements of Mn concentrations originating from both anthropogenic and dust sources can lead to substantial impacts of the water-soluble fraction of Mn on adverse health effects (i.e., ROS production) in the East Asian outflow region (e.g., Nishita-Hara et al., 2019) and ocean biogeochemistry as a micronutrient to limit or co-limit microbial activities (e.g., Ahlgren et al., 2014; Moore et al., 2013). Further characterizations of the $f_{sol}$ of Mn are critically needed to elucidate their variabilities and differences between anthropogenic and dust sources.



## 4 Conclusions

In this study, semi-continuous measurements of the elemental composition of fine mode (PM$_{2.5}$) aerosols were conducted on Fukue Island in western Japan during the spring of 2018. An automated X-ray fluorescence analyzer, PX-375, was successfully deployed, which has the capability to measure the aerosol's elemental compositions with a temporal resolution of a few hours even in remote regions after long-range transport from emission source regions. Four hourly mass concentrations of climatologically and geochemically important elements, such as S, Pb, Cu, Si, Ca, Fe, and Mn, in PM$_{2.5}$ aerosols were reported during the observation period. Furthermore, measurements of other atmospheric compositions, such as BC and CO, were conducted to analyze the emission and transport from the combustion sources over the East Asian continent. Positive correlations of Pb and Cu with BC and CO were found during the observation period, which indicate that the emission sources of these metals share the region where the large CO (and BC) emission sources are located. The use of CO, which is physicochemically stable on the time scale of atmospheric transport of aerosol particles, enabled us, for the first time, to successfully analyze the emissions of Pb and Cu and the impacts of the wet removal during the transport on their concentrations through the observations at the receptor site. The continental outflow air masses with minimized impacts of the wet removal during the transport were extracted to elucidate the emission ratio of Pb and Cu to CO, which were evaluated for the spring of 2018 in the East Asian outflow to be 152.7 and 63.1 µg g$^{-1}$, respectively. With increasing APT, the transport efficiency of Pb was found to decrease similar to that of BC, suggesting that the wet removal rates of Pb were almost the same as those of BC during the springtime East Asian outflow. Notably, Cu possessed a nature different from BC and Pb in terms of emission sources and wet removal in the East Asian outflow. Further simultaneous time-series characterizations of such elemental concentrations and mixing states are highly desirable to elucidate the mechanisms controlling their wet removal rates. The source apportionment of these elements was evaluated by comparing the simulations obtained from the IMPACT model with those observed by the PX-375 and was verified in terms of the emission and wet removal processes. From the analysis of the tagged tracer simulations by the IMPACT model, BC and Si could be used as tracers for anthropogenic and dust emissions, respectively, during the observation period. The source apportionments of Fe and Mn in PM$_{2.5}$ aerosols were found based on MLR analysis that used observed Si and BC tracers, and revealed that the anthropogenic contribution to total Fe and Mn in PM$_{2.5}$ aerosols was 17% and 44%, respectively, on average in the spring of 2018, corresponding to the range of 2%−29% and 9%−68% during the high PM$_{2.5}$ concentration periods depending on their air mass origins. However, the anthropogenic contributions of Fe were not dominant; the variations of anthropogenic Fe can regulate adverse health effects and ocean biogeochemistry due to its higher water solubility. The modeled BC, Pb, Cu, and Fe were evaluated by separately considering their emission and transport. Ratios of modeled to observed concentrations for these components were analyzed in terms of the APT from the East Asian continent. This indicated that the current model simulations overestimated the CEDS (v2021-02-05)-based emissions of BC by 44% and underestimated Cu by 45%, anthropogenic Fe by 28% in East Asia, and the wet deposition rates for BC and Pb. This study provides insights into the importance of removal process investigations, source apportionment of Fe, and high-temporal resolution measurements of aerosol trace elemental compositions.



**Data availability**

All the observation and model simulation data used in this study except the lidar observation data sets will be available at a data repository such as Zenodo and PANGAEA. The lidar observation data sets are available at the website of AD-Net, hosted by the National Institute for Environmental Studies, Japan (https://www-lidar.nies.go.jp/AD-Net/).

**Author contribution**

**Conceptualization:** T. M.; **Data curation:** T. M., A. I., A. S.; **Formal analysis:** T. M.; **Funding acquisition:** A. I., T. M.; **Investigation:** All authors; **Methodology:** T. M., A. I.; **Writing – original draft preparation:** T. M.; **Writing – review & editing:** All authors

**Competing interests**

Y. K. is an editorial board member of *Atmospheric Chemistry and Physics*. Y. M. and E. M. are employees of Horiba Ltd., which developed the continuous particulate monitor with the X-ray fluorescence analysis (PX-375) used in this study.

**Acknowledgements**

This study was supported by the JSPS KAKENHI (Grant No.: JP20H04329), MEXT-Program for the Advanced Studies of Climate Change Projection (SENTAN) (Grant No.: JPMXD0722681344), Environmental Research and Technology Development Fund (Grant no.: JPMEERF20222001) of the Environmental Restoration and Conservation Agency provided by Ministry of the Environment of Japan, and Steel Foundation for Environmental Protection Technology (Grant No.: C-48-12). We acknowledge M. Kubo, T. Takamura, and H. Irie (Chiba University) for their support at the Fukue Island's Atmospheric Environment Monitoring Station. Numerical simulations were performed using the Earth Simulator 4 at the Japan Agency for Marine-Earth Science and Technology. We would like to thank Editage (https://www.editage.com/) for English language editing.

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



**Figure 1.** Temporal variations of concentrations of (a) PM$_{2.5}$ (black) and sulfate (red), (b) black carbon (BC) (black), Pb (magenta), Cu (blue), and carbon monoxide (CO) (gray), (c) Si (brown), Ca (orange), and Fe (black). The selected periods when the Japanese short-term AQS for PM$_{2.5}$ was violated (daily PM$_{2.5}$ >35 μg m$^{-3}$) are highlighted by the shaded areas. (d, e, and f) 5-d backward trajectories for the selected periods P1 (March 25–26, 2018), P2 (April 15–18, 2018), and P3 (April 29, 2018).










**Figure 2.** Concentration-weighted trajectories of PM$_{2.5}$, sulfate (SO$_4^{2-}$), Pb, Cu, black carbon (BC), and Fe during the study period. Color scales are adjusted to the same range of 10%–90% of their concentration values for all the plots. The location of Fukue Island is indicated by open circles.





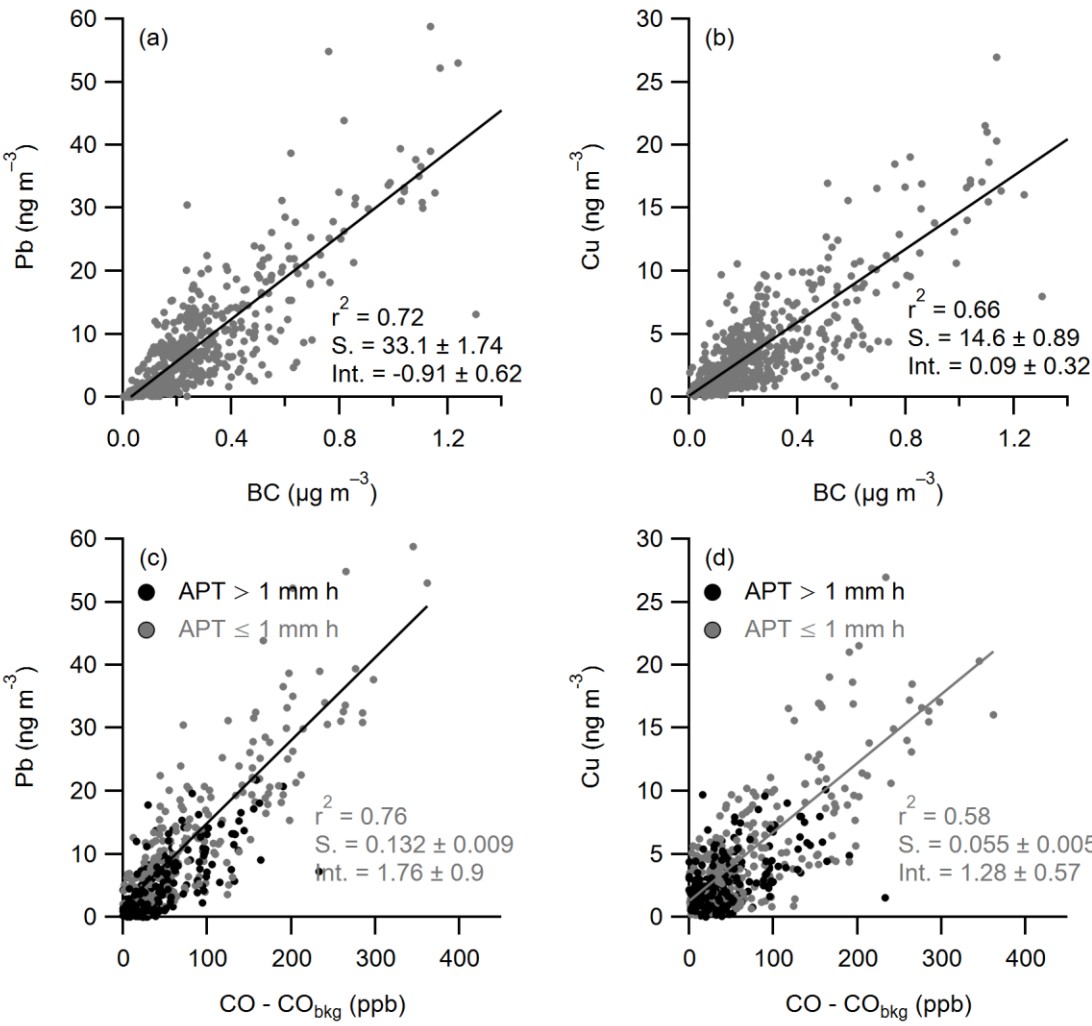

**Figure 3.** Correlations of black carbon (BC) with Pb (a) and Cu (b), and the enhancement of carbon monoxide above the background ($\Delta CO \equiv CO - CO_{bkg}$) with Pb (c) and Cu (d). In (c) and (d), the data points with the accumulated precipitation along trajectories (APT) higher than 1 mm h are highlighted as black circles.



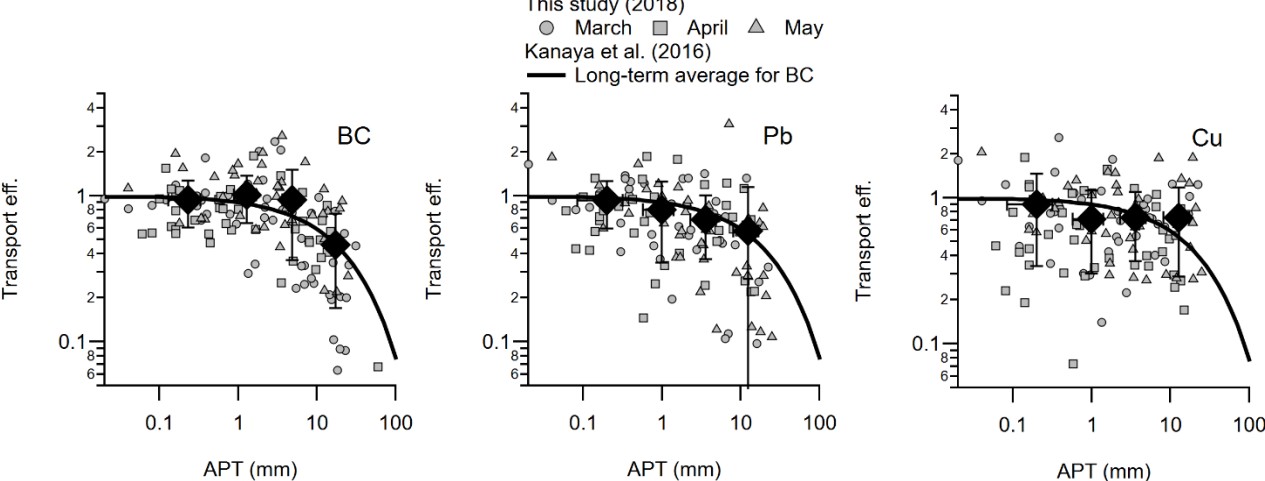

**Figure 4.** Transport efficiency for black carbon (BC) (left), Pb (middle), and Cu (right) as a function of accumulated precipitation along trajectories (APT) during the observation period. Shaded markers for 4-hourly data points are differentiated by the month (circles (March), squares (April), and triangles (May)). Filled black markers and error bars represent the binned averages and standard deviations. As a reference of the long-term tendency, the relationship between $TE_{BC}$ and APT in East Asia evaluated by our previous study (Kanaya et al., 2016) is overlayed in all the figures (solid lines).


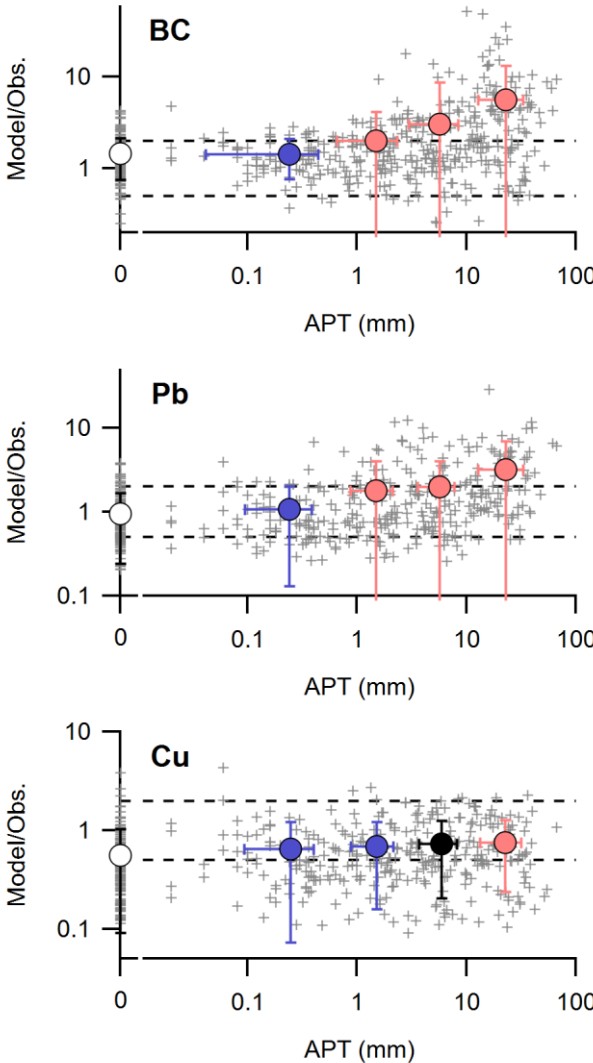

**Figure 5.** Model/Observed (M/O) ratios for black carbon (BC) (top), Pb (middle), and Cu (bottom) as a function of the accumulated precipitation along trajectories (APT) during the observation period. Shaded cross markers represent all 4-hourly data points with data higher than the limits of detection of BC, Pb, and Cu. Circles with the error bars are binned averages and standard deviations and are colored by the statistical significance of the differences in the M/O ratios from that at APT of 0 (student's t test results of $p < 0.01$ (red), $0.01 < p < 0.05$ (black), and $p > 0.05$ (blue)). The dashed lines depict the boundaries of factor of 2.





**Figure 6.** (a) Temporal variations of the observed $PM_{2.5}$ (black) and estimated BC-related $PM_{2.5}$ (blue), sulfate-related $PM_{2.5}$ (red), and dust-related $PM_{2.5}$ (brown) in Fukue Island. The selected periods when the Japanese air quality standard for the short-term (daily $PM_{2.5} > 35$ μg m$^{-3}$) was violated are highlighted by shaded areas. (b) Average observed concentrations of $PM_{2.5}$ (black bars) and those reconstructed using the tracers (colored bars: BC-related (blue), sulfate-related (red), dust (brown), and background (white)) for the entire period and selected periods (P1, P2, and P3). The error bars are standard deviations. (c) Temporal variations of the observed total extinction coefficient (black) and classified dust extinction coefficient (brown) at the altitude of 120−240 m in Fukue Island. (d) Correlation between the estimated contribution of dust to $PM_{2.5}$ mass and the lidar-derived contribution of dust to total extinction at the altitude of 120−240m (black for the selected periods (P1, P2, and P3) and shaded for other periods).







**Figure 7.** (a) Temporal variations of the observed Fe (black), and estimated anthropogenic Fe (blue), and dust Fe (brown) in Fukue Island. (b) Average observed concentrations of Fe (black bars) and those reconstructed using the tracers (colored bars: anthropogenic (blue) and dust (brown)) for the entire period and selected periods (P1, P2, and P3). The blue open markers and dashed lines are the estimated anthropogenic contribution to total Fe for the entire period and selected periods (P1, P2, and P3), respectively. (c) Temporal variations of the observed Mn (black), estimated anthropogenic Mn (blue), and dust Mn (brown) in Fukue Island. (d) Average observed concentrations of Mn (black bars) and those reconstructed using the tracers (colored bars: anthropogenic (blue) and dust (brown)) for the entire period and selected periods (P1, P2, and P3). The blue open markers and dashed lines are the estimated anthropogenic contribution to total Mn for the entire period and selected periods (P1, P2, and P3), respectively. The selected periods when the Japanese air quality standard for the short-term (daily $PM_{2.5} > 35$ µg m$^{-3}$) was violated are highlighted by shaded areas in (a) and (b). The error bars in (c) and (d) are standard deviations for the concentrations and the anthropogenic contributions.



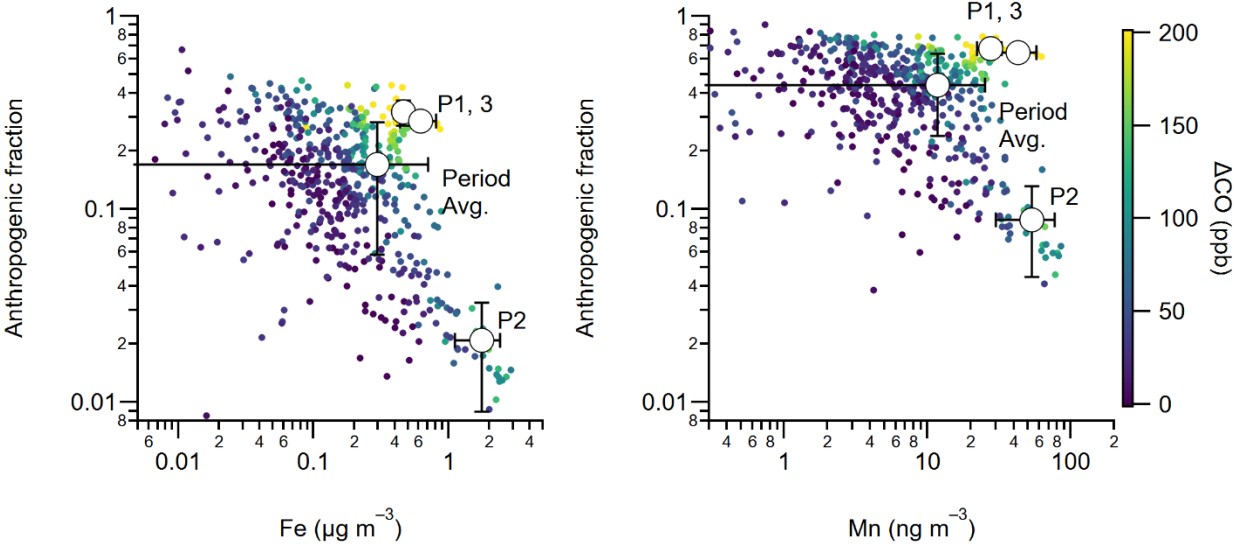

**Figure 8.** Relationship between the observed concentrations of Fe (left) and Mn (right) and their estimated anthropogenic
contributions. The data points are colored by the values of the enhancement of carbon monoxide above the background ($\Delta CO$
$\equiv CO - CO_{bkg}$). The open markers and error bars in the figure show the averages, standard deviations of the concentrations,
and the anthropogenic contributions of Fe and Mn for the entire period and selected periods (P1, P2, and P3).