# Peer review of "Trace elements in $PM_{2.5}$ aerosols in East Asian outflow in the spring of 2018: Emission, transport, and source apportionment"

_EGUsphere, 2023_

## Author Comment (AC1)

**Response to the comments of Reviewer #1:**

**General comments:**

**The authors present the field observation data of atmospheric trace components observed at a remote island in Japan and analyze the data using a backward trajectory technique, a three-dimensional model, and source apportionment based on a multiple linear regression method. Materials presented in the manuscript are interesting and well suited to the scope of the current journal. The manuscript is well written, and the logic is fine. The manuscript will be accepted in the journal after the authors revise the manuscript by reflecting the following general and specific comments.**

We appreciate the reviewer's helpful and constructive comments on the manuscript entitled "Trace elements in PM$_{2.5}$ aerosols in East Asian outflow in the spring of 2018: Emission, transport, and source apportionment".  As the reviewers suggested, we have modified the manuscript.  Major points for the revisions are listed as follows.

1) The improvements on the model predictions of Cu concentrations were added.  Considering the Cu smelting as an additional source of Cu in the improved simulation allowed us to correct the underestimation of Cu concentrations in the base simulation.

2) In relation to 1), new figures were added to the supplement information (Figure S6 and S7 in the revised SI) and some figures shown in the original manuscript were modified.

*Note that the authors' replies were in red.

1. The authors use the term "anthropogenic sources", but it can be divided into combustion and non-combustion sources and this separation might be the key to the study. BC and CO are mainly coming from the combustion sources, and probably so as Pb, but may not the case for Cu. Could you stress more on the existence and impacts of non-combustion sources of elements for your analysis? For instance, does the IMPACT model consider non-combustion sources for Pb and Cu? If not, can it be a cause of underestimation of simulated Cu at the observation site? If the authors neglected the contribution of elements from non-combustion sources because the sizes of aerosols are larger than PM$_5$ and thus the out of scope of the study, please mention this in the manuscript. There may be certain levels of non-combustion origins for anthropogenic Fe, too.

The mass concentrations of Cu-particles larger than 2.5 µm were simulated by the IMPACT model (e.g., Ito and Miyakawa, 2023), but the long-range transport of such coarse particles was probably not effective to account for the major part of the Cu concentrations observed at the observation site because of their short lifetime.  And the observations of elemental compositions were based on the PM$_{2.5}$ size cut (see Ln. 112–113 in section 2.1) and were compared with the

modeled elemental concentrations of aerosols smaller than 2.5 µm (see Ln. 175–177 in section 2.3). Therefore, the characterizations of coarse mode particles were out of scope of this study. The base version of IMPACT model considers the non-combustion (metal smelting) source of only Fe. To clarify how other elements than Fe from the metal production sector (smelting) were considered in the base version of the IMPACT model, we added the following sentence at the last of section 2.3.

"The simulation with the low estimate of smelting Fe emission factors showed the best agreement with field data, suggesting efficient and effective air quality management control strategies for the smelting facilities (Ito and Miyakawa, 2023). Thus, we did not consider the smelting emissions for other elements in the base version of the IMPACT model."

Nevertheless, the filtering units have only a certain efficiency, leading to an unavoidable release of air pollutants from the smelting facilities (e.g., Barcan, 2002; Sorooshian et al., 2012). As suggested, this can be a cause of the underestimation of the model-derived Cu concentrations. Given the large uncertainties in Cu-containing aerosols emitted from the smelting source, the underestimation of Cu concentrations in the model simulations may be corrected by including the contributions of the smelting source. We analyzed the model-to-observation differences for Cu with additional Cu emitted from the Cu smelting sources ($Cu_{CuSmelt}$). In this study, we calculated the averages of Cu content ($11.35 \pm 6.53$ wt.%) and Fe content ($1.37 \pm 0.47$ wt.%) in stack particulates emitted from three Cu smelters (Skeaff et al., 2011), and used their ratio, $(Cu/Fe)_{CuSmelt}$ of 8.31 wt.% wt.%$^{-1}$, as a scaling factor to estimate the $Cu_{CuSmelt}$ concentrations using the Fe emissions from the Cu smelting sources. It should be noted that $(Cu/Fe)_{CuSmelt}$ can be highly variable depending on the phase of processes and the types of the smelting facilities (e.g., Barcan, 2002; Skeaff et al., 2011) and size-dependent (Sorooshian et al., 2012), the latter of which showed the smaller values of $(Cu/Fe)_{CuSmelt}$ for submicron particles than supermicron particles. Here, the underestimated part of Cu concentrations in the base simulation was assumed to be those from the Cu smelting source and was compared with the $Cu_{CuSmelt}$ in the following figure (Figure S6 which is newly prepared). Although the correlation for 4-hourly data set is scattered ($r^2 = 0.22$), the bias can be mostly accounted for by the Cu smelting sources. A new figure for the TE-APT relationship for the observed and modeled concentrations of BC, Pb, Cu (base simulation), and Cu (improved simulation) was prepared (Figure S7). The relevant figures were revised by including the modeled Cu concentrations in the improved simulation. (Figures 5, S5, and S8).
We added the following sentences to the last of section 2.3, to describe how the $Cu_{CuSmelt}$ concentrations was estimated.

"Nevertheless, the filtering units only have a certain efficiency, leading to an unavoidable release of air pollutants from the smelting facilities (e.g., Barcan, 2002; Sorooshian et al., 2012). In this study, the concentrations of Cu emitted from the Cu smelting sources ($Cu_{CuSmelt}$) were estimated in the following way.  We calculated the averages of Cu content ($11.35 \pm 6.53$ wt.%) and Fe content ($1.37 \pm 0.47$ wt.%) in stack particulates emitted from three Cu smelters (Skeaff et al., 2011), and then used their ratio, $(Cu/Fe)_{CuSmelt}$, of 8.31 wt.% wt.%$^{-1}$ as a scaling factor to estimate the $Cu_{CuSmelt}$ concentrations using the Fe emissions from the Cu smelting sources in the improved version of the IMPACT model."

Another non-combustion source, the resuspended road dust from mechanically generated tire or brake wear, is not explicitly simulated by the IMPACT model.  Because brake wear Cu-containing particles can exist in fine mode (i.e., $PM_{2.5}$) (e.g., Hagino et al., 2016), this is a possible cause of the underestimation of the modeled Cu concentrations.  We added the following sentence behind the descriptions about the estimation of $Cu_{CuSmelt}$ concentrations.

"Note that the resuspended road dust from mechanically generated tire or brake wear, which can be an important non-combustion source of Cu, is not explicitly simulated by the IMPACT model."

Then, we modified the last paragraph of section 3.4, to describe the comparison of the underestimated fraction of Cu in the base simulations with $Cu_{CuSmelt}$ and the other cause for the model underestimation (i.e., the impact of the road brake).

[revised manuscript text omitted]

**Specific comments:**
*Abstract:*
**2. Ln. 10: "impact ocean biogeochemistry" -> "impact human health and ocean biogeochemistry", because the authors mentioned the impact of human health as well, in the latter part of abstract (Ln. 26) and in the Introduction section (First paragraph, regarding reactive oxygen species).**
Revised as suggested.

"impact Earth's radiative budget, human health, and ocean biogeochemistry"

**3. Ln. 13: "S" is not mentioned in the abstract and rather "Mn" may be an important element in this study. Please consider including Mn in the sentence, at least. It is up to authors' decision whether to exclude S from the sentence, though.**
We replaced "S" in this sentence by "Mn", as suggested.

"Temporal variations in mass concentrations of geochemically important elements for this period, such as Pb, Cu, Si, Fe, and Mn, and their relationships with the emission tracers, carbon monoxide (CO) and black carbon (BC), were reported."

**4. Ln. 20: It is relating to the major comment #1, but I am not sure whether derivations of Pb/CO and Cu/CO ratios are meaningful or not, because numerators (Pb and Cu) may come from both non-combustion and combustion sources, while denominator only comes from combustion sources.**

In this study, CO was analyzed as a tracer compound emitted from the areas where the major anthropogenic emission sources located. As pointed out in General comment 1, metallic species can be emitted from non-combustion sources. However, the geographical distributions of combustion sources are not so different from those of non-combustion sources, as indicated by the CWT analyses (see Fig. 2). Furthermore, the correlations between CO and ozone ($O_3$) in continental outflows have been analyzed to diagnose the photochemical production of $O_3$ (e.g., Parrish et al., 1993), although $O_3$ was not directly emitted from the combustion sources. CO-tracer method can provide a "top-down" estimate of the emission ratio of the targeted species to CO over the anthropogenic emission regions.

"Positive correlations of Pb and Cu with BC and CO and the similarity of their concentration-weight trajectories indicated that the emission sources of these metals share the region where the large CO (and BC) emission sources are located and that CO can be regarded as a tracer of continental anthropogenic emissions. The air masses with minimized impacts of the wet removal during transport were extracted to elucidate the "top-down" emission ratio of Pb and Cu to CO, which were, for the first time, evaluated as 152.7 and 63.1 $\mu g\ g^{-1}$, respectively, during the spring of 2018 in the East Asian outflow."

*Material and methods:*
**5. Ln. 119: "15 and 50 kV". Why are two different voltage levels used in the analysis?**
In order to acquire the characteristic X-ray emissions from specific elements, the irradiations of electrons accelerated at the voltage (at least two times) higher than their critical excitation energies Ec.  For example, the Ec of Fe is 7.11 kV (for K edge), indicating the need to apply at least around 15 kV for the detection of the characteristic X-ray emission from Fe with high sensitivity.  In general, the higher accelerating voltage (e.g., 50 kV) is needed to detect the heavier elements (e.g., Pb).  It should be noted that too high energies compared to the required excitation energy led to the uncertainties caused by self-absorption of X-ray fluorescence.  The voltage applied for the analysis to target elements needs to be properly selected.  We hence selected two voltages for the XRF analysis in the PX-375.

**6. Ln. 128: Please explain more about the "uncertainties" here. What are the exact measures for the values? Are they normalized errors? Are they the uncertainties of PX-375 data against the reference data, that are measured by IC and ICP-MS? Or are they relative errors between PX-375 and IC/ICP-MS?**
The uncertainties were evaluated by considering the intercomparison between the XRF analyses by the PX-375 and the reference analyses.  The linear regression slopes were regarded a

measure of the accuracy and the ratio of 95% confidence intervals to the average of the slopes were regarded a measure of the precision.  The uncertainties estimated in the original manuscript were evaluated by combining both (i.e., root-sum-square).  Therefore, the uncertainties were evaluated relative to the reference analyses and cloud be rephrased by the relative errors.  The sentence was revised as follows.

"Based on the intercomparison results (linear regression slopes and their variances), the relative errors in the on-site measurements of the concentrations of Cl, S, Fe, Pb, Mn, Cu, K, and Ca were evaluated to be 66%, 6%, 26%, 15%, 23%, 30%, 28%, and 33%, respectively"

**7. Lns. 142-145: Please include time period also. "(6600)" in Ln. 156 may be the number of trajectories but I have no idea why the total number is 6600. Time resolution of trajectory is hourly, so 24 (hrs) x 3 (layers) x 90 (days) = 6480, a little bit different from 6600, but anyway the same order. However, as written later in Lns. 157-158, L in Eq. 1 is based on 4 hourly values to match with the time resolution of PX-375, so that the orders of L may be smaller, around 1620, right?**

We calculated the backward trajectories from 10:00 LT March 1, 2018 to 00:00 LT June 1, 2018 (not 90 days).  By also considering the missing data of the elemental concentrations, total number is determined to be 6600.  As described in the sentence at line 157–158 of the original manuscript, the same concentrations were used for a 4 hour duration in the CWT calculation process.  This was intended to use as many trajectories as possible in the CWT calculation.

*Results and Discussion*

**8. Lns. 190-191, "with small Japanese emission impacts": Fig. S2 indicates the residence time of trajectories and does not tell the impact of emissions. The impacts of emission depend on emission flux and distance from the source. Please rephrase the relevant sentence by what Fig. S2 really tells.**

As pointed out, the information of the residence time is not sufficient to judge the quantitative impacts of "emissions".  The sentence was revised as follows.

"From March to May 2018, the air masses were frequently transported directly from the East Asian continent (not via mainland Japan) to the observation site on Fukue Island (Figure S2)."

**9. Lns. 192-193: It is not clear why the authors present Fig. S3. Probably "Notably, air masses … during the observation period." is the reason why, but some more words may be needed to make the readers compelling. Please add some more words to explain why the**

**fact that no correlation is found between residence time over the continent and APT in Fig. S3 is important for the analysis (and for which analysis?) of this study.**

We prepared this figure to show no significant bias of air mass origins and pathways with respect to the data selection of air masses with or without wet depositions (non-zero or zero APT, respectively). As suggested, it was not clear in the original manuscript. We modified the sentence at line 192–193 ("Notably, ~ observation period") as follows.

"Thus, there was no significant bias of air mass origins and pathways concerning the data selection of air masses with non-zero or zero APT."

**10. Lns. 243-245. "rainout" means in-cloud scavenging, right? "wet depositions" includes both in-cloud and below-cloud maybe. Do the authors intend to mean that the deposition mechanisms of BC and Pb/Cu are different? Or the same (both removed by in-cloud and/or below-cloud scavenging)? Anyway, please explain why the authors assume so? Is it because mixing-state and sizes of BC, Pb, and Cu are different (or similar) with each other?**

In this sentence, rainout process (in-cloud scavenging and the subsequent precipitation) was referred because this process was identified in previous studies as the major removal process during the transport in East Asian outflow (Kanaya et al., 2016; Miyakawa et al., 2017; Moteki et al., 2012). In general, all the fine aerosols can be removed more efficiently by the rainout than by washout (below-cloud scavenging). We modified the sentence as follows.

"Because BC aerosol particles in the East Asian outflow are efficiently removed from the atmosphere by the in-cloud scavenging and the subsequent precipitation (Kanaya et al., 2016; Miyakawa et al., 2017; Moteki et al., 2012), Pb and Cu can also be affected by these processes during transport."

**11. Ln. 274, Fig. 4: Please reconfirm the unit of APT. It was "mm h" in Fig. 3 as well as main text, while "mm" here. Please also check it in Figs. 5, and S10, and elsewhere, if any.**

"mm" is correct. We modified as suggested.

**12. Ln. 277: "Cu has characteristics of emission and mixing states different from BC and Pb". This is interesting and can be an answer for my comments #1 and #10. Not only "emission" and "mixing state", but also "size" may be an important factor to affect wet removal and thus determine the transport efficiency, but why is it not included? Is "size" already included in "emission" or "mixing state"?**

"Mixing states" in the original manuscript does not explicitly include "size" information. As pointed out, size can be important even though all the measurements of aerosols compositions were performed with the same aerodynamic size cut (i.e., $PM_{2.5}$ in this study). In previous studies at urban cities, the mass size distributions of Cu had the different shape from those of other components such as Pb and elemental carbon (e.g., Fang et al., 2017; Yang et al., 2023). We modified the sentence as follows.

"While $TE_{Pb}$ showed a similar relationship with the APT to $TE_{BC}$, the decreasing trend of the $TE_{Cu}$-APT relationship was not steep, probably because Cu has characteristics of emission, size distributions, and mixing states different from BC and Pb. Previous studies in urban cities illustrated that Cu exhibited different size distributions from other components such as Pb (e.g., Yang et al., 2023) and BC (e.g., Fang et al., 2017). Kinase et al. (2022) reported continuous observations of trace metals in $PM_{2.5}$ aerosols in an urban area of western Japan and showed that Cu concentrations were poorly correlated ($r^2 = 0.17$) with Pb concentrations when the local impact was significant and that approximately 60% of Cu-containing particles did not contain Pb as analyzed using an electron microscopic technique."

**13. Ln. 295: "the removal processes and emissions of Cu were not properly simulated by the IMPACT model". I have an opposite impression from the authors for what Fig. 5 tells: constant Model/Obs ratio of Cu for different APT regions means wet removal processes of Cu in the model were rather successful! Could you explain more about the difference of wet removal calculations for BC, Pb, and Cu in the IMPACT model? (It should already be written in the description paper, Ito and Miyakawa, 2023, but please explain here again, because trends of BC/Pb and Cu are remarkably different in Fig. 5).**
Currently, IMPACT model simulates the wet removal processes of aerosol particles through the in-cloud (e.g., Ito and Xu, 2014) and below-cloud scavenging (e.g., Ito and Kok, 2017). All the metal components are removed as the same rates as BC (no differences in the removal efficiencies among elements in anthropogenic aerosols) in the model simulations, assuming internal mixing of BC and trace elements (e.g., Ito and Feng, 2010). We added this point at the last paragraph in section 2.3.

"The IMPACT model simulated the wet removal processes through the in-cloud (e.g., Ito and Xu, 2014) and below-cloud scavenging (e.g., Ito and Kok, 2017). All the metal components are removed at the same rates as BC (i.e., no differences in the removal efficiencies among elements in anthropogenic aerosols) in the model simulations, assuming internal mixing of BC and trace elements (e.g., Ito and Feng, 2010), which will be discussed in the later sections."

Major cause of the discrepancy between observed and modeled Cu concentrations (around 50%) can be the uncertainty with respect to the emissions, as discussed in the response to the reviewer's general comment. However, Fig. S6 in the original SI suggested that the transport efficiency of Cu also affected the model-to-observation ratios of Cu (i.e., anticorrelation). We found that (1) the base emissions missed the contributions from the Cu smelting sources, (2) the transport (or removal) efficiency of Cu was not well accounted for by the APT but was in relation to the model underestimations, and (3) the cause of (2) can be the differences in the particle mixing state and size distributions between Cu and other components such as BC and Pb as indicated by some previous studies (e.g., Fang et al., 2017; Kinase et al., 2022; Yang et al., 2023). These are also related to the response to the reviewer comment 12. We modified the sentence Ln 295–296 as follows.

"Although the APT was not a strong forcing factor of $TE_{Cu}$, the removal processes of Cu were not properly simulated by the IMPACT model, as indicated by the relationship between the M/O ratios and TE for BC, Pb, and Cu, as shown in Figure S8. We found that the wet removal of aerosols in the IMPACT model needs to be revised."

**14. Ln. 304, Fig. S7: Are they correlations for observations or IMPACT? The panels look correlations for observation data, but from the main text, they might be the data of IMPACT, because the relevant paragraph in the main text mentions IMPACT in the beginning. Please specify.**

All these data are from the observation. We revised the sentences L302–305 and the figure caption of Fig. S7 (S9) in the original (revised) SI as follows.

"Source apportionment of $PM_{2.5}$ aerosols was conducted using the observed BC, Si, and sulfate concentrations ($[BC]_t$, $[Si]_t$, and $[SO_4^{2-}]_t$, respectively) to validate the use of these tracers for the source apportionment of Fe and Mn. Because $[BC]_t$ showed a positive correlation with $[SO_4^{2-}]_t$ (**Figure S9**), the temporal variations of BC may not be completely independent from those of sulfate."

"Figure S9. Observed correlations of (a) black carbon (BC) and sulfate concentrations, (b) Si and sulfate concentrations, (c) BC and Si concentrations, (d) the enhancements of carbon monoxide (CO) from the background ($\Delta$CO, see the main texts for the details) and BC concentrations, and (e) $\Delta$CO and sulfate concentrations."

**15. Ln. 357: "dust-Mn concentrations were underestimated by the IMPACT model". Why could it happen, although the simulated dust-Fe is successful? Dust-Mn and dust-Fe are simulated using the same total dust mass concentrations, right? I mean, dust-Mn and dust-Fe are derived from the common dust emission scheme. Is it because the Mn-content set in the model (global scale?) is very different from that in reality (Asian dust)? How are the Fe and Mn contents in the model different (or similar) from NIES CRM NO. 30 Gobi Kosa Dust, for example?**

The major reason why the IMPACT model reasonably well predicted dust-Fe concentrations is that the IMPACT model adjusted the Fe content for each Fe species in clay-sized soils for East Asian dust aerosols (3.83% on average) based on the Fe content for each Fe species in the clay-sized fraction of Chines desert sediment (see Ito and Miyakawa, 2023; Lu et al., 2017; and references therein). On the other hand, the IMPACT model prescribed Mn content in dust aerosols from the NIES CRM No. 30 Gobi Kosa (0.077wt%). In the observation-based studies (Jeong, 2020 and Wang et al., 2011), Mn contents were evaluated to be higher (~0.11wt%) than prescribed in the model. The prescribed value of Mn may be lower than the real one. New estimation in this study (0.21wt%) was 2–3 times higher than both prescribed in the model and those evaluated in the previous studies, indicating the possibility to overestimate dust-Mn concentrations. These are the possible reasons why the IMPACT model underestimated the dust-Mn concentrations and were described in section 3.5.4 of the original manuscript.
We added the following sentence at the last of section 3.5.2.

"The underestimation of anthropogenic- and dust-Mn concentrations can be accounted for by the uncertainties in the anthropogenic emissions of Mn and the Mn content in dust aerosols (described in the later section) prescribed in the IMPACT model."

**16. Lns. 383-386: I am wondering if the denominators in Fig. S11 of all studies (Jeong, Wang, and IMPACT) are the same. Are they really PM$_{2.5}$-dust only or total PM$_{2.5}$ concentrations during the dust events, which includes components other than dust. The simulation may be the former, but for observations could be the latter.**

As pointed out, Jeong (2020) and Wang et al. (2011) analyzed the aerosol samples collected in Korea and China during "strong" dust events, whereas Ito and Miyakawa (2023) simulated and prescribed elemental compositions of dust aerosols. We selected these references to compare our multiple linear regression (MLR) result with different methods. As suggested, the observation-based results can be affected by the components other than dust. However, in their studies, the observed concentrations of aerosols were higher than ~150 µg m$^{-3}$, and the enrichment factors of Fe, and Mn were 1–2, which was comparable to those for Gobi Desert soil

(Jeong, 2020).  This indicated that the effects of non-dust components on the dust elemental composition characterizations were probably insignificant.  We added the following sentence behind the first sentence of section 3.5.4.

"Note that these observation-based characterizations of dust elemental compositions were not significantly affected by non-dust components as expected from the enrichment factor analyses (Jeong, 2020; Wang et al., 2011)."

*Conclusions*

**17. Lns. 457-458, "such as elemental concentrations and mixing states". Probably size distribution is also important, as commented in #12. Or does the term "mixing state" include size information as well?**

We agreed with the reviewer's comment to this point.  The sentence was modified as follows.

"Further simultaneous time-series characterizations of size-resolved elemental concentrations and mixing states are highly desirable to elucidate the mechanisms controlling their wet removal rates."

*Supporting information*

**18. Sect. S1: $Cl^-$ is used for the contribution of sea-salt particles but it is evaporative so $Na^+$ may be a better indicator. $Na^+$ may be difficult to be analyzed by ICP-MS or PX-375, but you have IC data, right? Why didn't you use IC $Na^+$ data for your analysis? This is also an additional comment on the main text, in Lns. 228-231, instead of $Ca^{2+}$, non-sea-salt $Ca^{2+}$ (derived by assuming $Na^+$ as fully originated from sea-salt) can be a better indicator for dust aerosols. (Certainly, you don't need nss-$Ca^{2+}$, as you have already Si as a good indicator)**

Our measurements of $PM_{2.5}$ elemental compositions using the PX375 allowed us to use the elemental Cl concentrations (not $Cl^-$).  We only have a very limited numbers of ionic concentration data from an IC technique applied to the collected filter spots.  This is because the IC analyses were performed to validate the performance of the PX-375 for the aerosols sampled before and after the observation period.  We therefore did not evaluate the concentrations of sea-salt (SS) from sodium ion concentrations and evaluated only the SS concentrations as an upper limit.  As the major purpose to estimate the concentration levels of sea-salt aerosols was to validate the MLR analysis for $PM_{2.5}$ total mass concentrations, evaluating the precise temporal variations of SS concentrations is of secondary importance.  Indeed, the data analysis using the equation S4 suggested that SS did not affect the source apportionment of $PM_{2.5}$ aerosols.

We just modified the sentence Ln33–35 as follows, to clarify that this is not a precise way to estimate the impact of SS.

"In this study, the possible impacts of SS aerosols on the source apportionment of $PM_{2.5}$ aerosols were assessed using the temporal variations of chlorine concentrations $[Cl]_t$ in $PM_{2.5}$ aerosols, which were measured using the PX-375."

**19. Caption of Fig. S8: what do you mean by "stacked"? (MPOA was stacked on the modeled SS).**

The modeled MPOA was stacked on the modeled SS to clearly show the total concentrations of sea-spraying aerosols (SSA). We modified the caption of Fig. S8 (S10) in the original (revised) SI as follows.

"The modeled marine primary organic aerosol (MPOA, light red area) concentrations were stacked on the modeled SS concentrations in (d) to illustrate the modeled total concentrations of sea-spraying aerosols (SSA) and the contributions of MPOA to total SSA."

**Reference papers newly included in the revised manuscript**

Barcan, V.: Nature and origin of multicomponent aerial emissions of the copper-nickel smelter complex. Environ. int., 28(6), 451–456, 2002.

Fang, T., Guo, H., Zeng, L., Verma, V., Nenes, A., and Weber, R. J.: Highly Acidic Ambient Particles, Soluble Metals, and Oxidative Potential: A Link between Sulfate and Aerosol Toxicity. Environ. Sci. Technol., 51(5), 2611–2620, https://doi.org/10.1021/acs.est.6b06151, 2017.

Hagino, H., Oyama, M., and Sasaki, S.: Laboratory testing of airborne brake wear particle emissions using a dynamometer system under urban city driving cycles. Atmos. environ., 131, 269-278, https://doi.org/10.1016/j.atmosenv.2016.02.014, 2016

Ito, A. and Feng, Y.: Role of dust alkalinity in acid mobilization of iron. Atmos. Chem. Phys., 10, 9237–9250, https://doi.org/10.5194/acp-10-9237-2010, 2010.

Ito, A. and Xu, L.: Response of acid mobilization of iron-containing mineral dust to improvement of air quality projected in the future. Atmos. Chem. Phys., 14, 3441–3459, https://doi.org/10.5194/acp-14-3441-2014, 2014.

Ito, A., and Kok, J. F.: Do dust emissions from sparsely vegetated regions dominate atmospheric iron supply to the Southern Ocean?. J. Geophys. Res. Atmos., 122, 3987–4002, https://doi.org/10.1002/2016JD025939, 2017.

Skeaff, J. M., Thibault, Y. and Hardy, D. J.: A new method for the characterisation and quantitative

speciation of base metal smelter stack particulates. Environ. Monit. Assess., 177, 165–192, https://doi.org/10.1007/s10661-010-1627-9, 2011.

Sorooshian, A., Csavina, J., Shingler, T., Dey, S., Brechtel, F. J., Saez, A. E., and Betterton, E. A.: Hygroscopic and Chemical Properties of Aerosols Collected near a Copper Smelter: Implications for Public and Environmental Health. Environ. Sci. Technol., 46(17), 9473–9480, https://doi.org/10.1021/es302275k, 2012.

Yang, J., Ma, L., He, X., Au, W. C., Miao, Y., Wang, W.-X., and Nah, T.: Measurement report: Abundance and fractional solubilities of aerosol metals in urban Hong Kong - insights into factors that control aerosol metal dissolution in an urban site in South China. Atmos. Chem. Phys., 23, 1403–1419, https://doi.org/10.5194/acp-23-1403-2023, 2023.

---

## Author Comment (AC2)

**Response to the comments of Reviewer#2:**

We appreciate the reviewer's helpful and constructive comments on the manuscript entitled "Trace elements in $PM_{2.5}$ aerosols in East Asian outflow in the spring of 2018: Emission, transport, and source apportionment". As the reviewers suggested, we have modified the manuscript. Major points for the revisions are listed as follows.

1) The improvements on the model predictions of Cu concentrations were added. Considering the Cu smelting as an additional source of Cu in the improved simulation allowed us to correct the underestimation of Cu concentrations in the base simulation.

2) In relation to 1), new figures were added to the supplement information (Figure S6 and S7 in the revised SI) and some figures shown in the original manuscript were modified.

*Note that the authors' replies were in red.

**Abstract:**

**Ln 10: 1st sentence can be changed to 'Trace metals in aerosol particles impact Earth's radiative balance, ocean biogeochemistry, and human health.' (as mentioned in the introduction). Also removing 'therefore' from the second sentence is suggested since it is not a conclusion for the previous one.**

We revised as suggested.

"Trace metals in aerosol particles impact Earth's radiative balance, human health, and ocean biogeochemistry. Semi-continuous measurements of the elemental composition of fine mode ($PM_{2.5}$) aerosols were conducted using an automated X-ray fluorescence analyzer on a remote island of Japan during the spring of 2018."

**Ln 26: Using the word 'Minor' for anthropogenic contribution of Fe is not recommended. Maybe using the term 'not dominant' like used in conclusions is better.**

We revised as suggested.

"However, despite the non-dominant anthropogenic contributions of Fe, they could adversely affect human health and ocean biogeochemistry owing to their higher water solubility."

**Introduction:**

**Ln 42: Change 'has been concerned' to 'has been a concern'.**

We revised as suggested.

"The oxidative potential of aerosol particles, which have the potential to generate ROS in cells

and cause oxidative stress to cells, has been a concern."

**Materials and methods:**

**Ln 118: Was the 4h analysis time period used so that sufficient mass could be collected on the tape, or for some other reason?**

We operated the PX-375 for 4 hours of the sampling and 4000 sec of the following XRF analysis in this study. As suggested, the measurement needs the enough amounts of aerosols on a spot on the filter tape. At the same time the shorter time resolution was preferable to compare the temporal variations of trace metals with those of tracer compounds (BC and CO).

**Ln 128: How were these uncertainties calculated?**

The uncertainties were evaluated by considering the intercomparison between the XRF analyses by the PX-375 and the reference analyses (IC and ICP-MS). The linear regression slopes were regarded a measure of the accuracy and the ratio of 95% confidence intervals to the average of the slopes were regarded a measure of the precision. The uncertainties estimated in the original manuscript were evaluated by combining both (i.e., root-sum-square). Therefore, the uncertainties were evaluated relative to the reference analyses and cloud be rephrased by the relative errors. The sentence was revised as follows.

"Based on the intercomparison results (linear regression slopes and their variances), the relative errors in the on-site measurements of the concentrations of Cl, S, Fe, Pb, Mn, Cu, K, and Ca were evaluated to be 66%, 6%, 26%, 15%, 23%, 30%, 28%, and 33%, respectively"

**Ln 146: Is there a particular reason why three-day APT was considered?**

We followed our previous studies (Kanaya et al., 2016; 2020; Miyakawa et al., 2017). Typical transport time from the East Asian continent to the observation site in Fukue island was ~2 days. To cover the period of the air mass transport from the source regions, we set 3 days for calculating the APT.

**Results:**

**Ln 250: States that 'Cu has different emission sources from BC, Pb and CO'. The correlations of Pb and Cu v/s BC/CO are not drastically different, so can that really imply a different emission source altogether?**

Figures 3c and 3d indicated only the differences in $r^2$ of the correlation with BC or CO between Pb and Cu, indicating the possibility of the differences in the emission sources. We modified the sentence as follows.

"This indicates that Cu possibly has different emission sources from BC, Pb, and CO in East Asia, even though the similarity of their geospatial patterns was indicated by the CWTs for BC, Pb, and Cu."

**Ln 690: Fig 4 and 5 give units of APT as mm, whereas Fig 3 gives it as mm h. Which is the correct one? Please check other figs in SI as well.**

"mm" is correct. We modified as suggested.

**Ln 295: Looking at Fig 5, it seems that Cu is simulated well by the IMPACT model, due to its consistent M/O across different APT ranges, but the text states otherwise. Can you elaborate more on this?**

As suggested, only looking at Fig. 5 let us find that M/O of Cu did not largely vary across different APT ranges. However, we found the relationship between M/O and transport efficiency of Cu, as shown in Figure S6 (S8) in the original (revised) SI. (This point has been described at the sentences Ln. 295–297 of the original manuscript.) We thus concluded that the APT was not a strong forcing factor to account for the model underestimations of Cu concentrations but the removal processes of aerosols (as seen in $TE_{BC}$, $TE_{Pb}$, and $TE_{Cu}$) were not properly simulated in the base and improved versions of the IMPACT model. We modified the sentence Ln. 295–296 as follows.

"Although the APT was not a strong forcing factor of $TE_{Cu}$, the removal processes of Cu were not properly simulated by the IMPACT model, as indicated by the relationship between the M/O ratios and TE for BC, Pb, and Cu, as shown in **Figure S8**. We found that the wet removal of aerosols in the IMPACT model need to be revised."

**Conclusions:**
**Ln 448: States that 'emission sources of these metals share the region where the large CO (and BC) emission sources are located.' I understand that this conclusion is based on CWT results. However, it is confusing to understand that Cu has a different emission source based on correlations but a similar region of emission. Please explain this part more clearly, and make edits to similar sentences made in the abstract and results too.**

The sentence at Ln 17–19 was modified as follows.

"Positive correlations of Pb and Cu with BC and CO and the similarity of their concentration-weight trajectories indicated that the emission sources of these metals share the region where the

large CO (and BC) emission sources are located and that CO can be regarded as a tracer of continental anthropogenic emissions."

The sentence at Ln 447–449 was modified as follows.

"Positive correlations of Pb and Cu with BC and CO and the similarity of their CWTs indicated that the emission sources of these metals share the region where the large CO (and BC) emission sources are located."

**Ln 460: BC is used as a tracer for anthropogenic emissions. What is novel about this part?**

BC is a well-known tracer for combustion sources (not only anthropogenic but also biomass burning (BB)). From the tagged tracer analysis in the IMPACT simulations, it was found in this study that BC mainly originated from anthropogenic combustion source (BB was not significant) in the East Asian outflow in the spring of 2018. To the best of our knowledge, for the first time, "the combination of BC and Si" in the multiple linear regression was used for the observation-based source apportionment of Fe and Mn.

**Reference papers newly included in the revised manuscript**

Barcan, V.: Nature and origin of multicomponent aerial emissions of the copper-nickel smelter complex. Environ. int., 28(6), 451–456, 2002.

Fang, T., Guo, H., Zeng, L., Verma, V., Nenes, A., and Weber, R. J.: Highly Acidic Ambient Particles, Soluble Metals, and Oxidative Potential: A Link between Sulfate and Aerosol Toxicity. Environ. Sci. Technol., 51(5), 2611–2620, https://doi.org/10.1021/acs.est.6b06151, 2017.

Hagino, H., Oyama, M., and Sasaki, S.: Laboratory testing of airborne brake wear particle emissions using a dynamometer system under urban city driving cycles. Atmos. environ., 131, 269-278, https://doi.org/10.1016/j.atmosenv.2016.02.014, 2016

Ito, A. and Feng, Y.: Role of dust alkalinity in acid mobilization of iron. Atmos. Chem. Phys., 10, 9237–9250, https://doi.org/10.5194/acp-10-9237-2010, 2010.

Ito, A. and Xu, L.: Response of acid mobilization of iron-containing mineral dust to improvement of air quality projected in the future. Atmos. Chem. Phys., 14, 3441–3459, https://doi.org/10.5194/acp-14-3441-2014, 2014.

Ito, A., and Kok, J. F.: Do dust emissions from sparsely vegetated regions dominate atmospheric iron supply to the Southern Ocean?. J. Geophys. Res. Atmos., 122, 3987–4002, https://doi.org/10.1002/2016JD025939, 2017.

Skeaff, J. M., Thibault, Y. and Hardy, D. J.: A new method for the characterisation and

quantitative speciation of base metal smelter stack particulates. Environ. Monit. Assess., 177, 165–192, https://doi.org/10.1007/s10661-010-1627-9, 2011.

Sorooshian, A., Csavina, J., Shingler, T., Dey, S., Brechtel, F. J., Saez, A. E., and Betterton, E. A.: Hygroscopic and Chemical Properties of Aerosols Collected near a Copper Smelter: Implications for Public and Environmental Health. Environ. Sci. Technol., 46(17), 9473–9480, https://doi.org/10.1021/es302275k, 2012.

Yang, J., Ma, L., He, X., Au, W. C., Miao, Y., Wang, W.-X., and Nah, T.: Measurement report: Abundance and fractional solubilities of aerosol metals in urban Hong Kong - insights into factors that control aerosol metal dissolution in an urban site in South China. Atmos. Chem. Phys., 23, 1403–1419, https://doi.org/10.5194/acp-23-1403-2023, 2023.